# Learning Coupled Continuous-Time Latent Dynamics from Irregular Events

Jiankai Zuo [1]   Yang Zhang[✉ 2]   Yu Zhang [3]   Jiarui Liang [4]   Yaying Zhang[✉ 4]

## Abstract

Modeling dynamic dependencies from irregularly sampled event sequences is a fundamental challenge in modern machine learning. In many real-world systems, individual-level states evolve continuously over time while being simultaneously influenced by population-level dynamics. However, existing methods typically model these processes in isolation or rely on discrete-time approximations that fail to capture long-range temporal irregularities and sparse observations. This paper studies the problem of learning coupled continuous-time latent dynamics from irregular events, where individual event sequences and global distributional processes evolve asynchronously and interact over time. We propose a Coupled Continuous-Time Latent Dynamics (CoCLD) framework that jointly models individual latent dynamics and population-level distributional shifts, and aligns them in a continuous-time latent space. CoCLD integrates a Diffusion-based Latent Interpolator with neural ordinary differential equations, enabling principled interpolation, generation, and alignment of latent states across arbitrary time points. We show that the proposed coupling mechanism yields a consistent estimator of continuous-time latent dynamics under sparse and irregular observations. Empirical evaluations show CoCLD effectively captures dynamic dependencies and generalizes across tasks like next-event prediction, mobility trajectory generation, and sequential behavior modeling, indicating that learning coupled continuous-time latent dynamics is a powerful paradigm for irregular event sequence modeling.

---

[1]School of Electronic and Information Engineering, Suzhou University of Science and Technology [2]The Anuradha and Vikas Sinha Department of Data Science, University of North Texas [3]School of Computing, Faculty of Science and Engineering, Macquarie University [4]School of Computer Science and Technology, Tongji University. Correspondence to: Yaying Zhang <yaying.zhang@tongji.edu.cn>, Yang Zhang <yang.zhang@unt.edu>.

*Proceedings of the 43rd International Conference on Machine Learning*, Seoul, South Korea. PMLR 306, 2026. Copyright 2026 by the author(s).

## 1. Introduction

Irregularly sampled event sequences (Chen et al., 2023a; Sun et al., 2024; Osin et al., 2025) arise throughout modern machine learning systems, including user interactions, mobility traces, check-in records, and online behavioral logs. In these settings, observations arrive at non-uniform, stochastic timestamps, often with long gaps and bursty activity. Such irregularity makes it difficult to learn faithful temporal dependencies: discrete-time models (Che et al., 2018) typically rely on fixed binning or step-wise updates that blur long-range temporal structure, while purely continuous-time models (Chen et al., 2018; Wang et al., 2025) may suffer when observations are extremely sparse, and the latent trajectory is weakly constrained between events.

A second, equally important challenge is that many real-world systems exhibit *multi-level dynamics*. Individual-level states (e.g., a user's preference or a vehicle's intent) evolve continuously, but they are also shaped by population-level distributional shifts (e.g., seasonal effects, global mobility patterns, platform-wide exposure changes). Conversely, aggregate distributions evolve partly due to collective individual behaviors. Despite their prevalence, capturing these coupled dynamics and bidirectional dependence remains non-trivial. Traditional sequence models, such as RNNs (Lu & Xu, 2024) and Transformers (Vaswani et al., 2017), typically rely on discrete-time approximations (Zhi et al., 2022) that fail to capture long-range temporal irregularities and sparse observations. In addition, most approaches (Yuan et al., 2022; Chan et al., 2023; Yu et al., 2024) model individual sequences and global trends in isolation or through late fusion, failing to capture their asynchronous yet tightly coupled evolution in continuous time.

Recent progress has highlighted two complementary directions. On one hand, the Neural Ordinary Differential Equations (Neural ODEs) (Chen et al., 2018; Li et al., 2025) provide principled continuous-time trajectories by learning vector fields and integrating them between observations, enabling modeling on irregular timestamps without discretization (Liu et al., 2025b). But they often struggle with extreme data sparsity—where observations are absent during critical intervals—and rarely account for the interactive evolution between individual trajectories and global trends. On the other hand, diffusion-based generative models (Wang et al.,

2023b; Fu et al., 2025) have demonstrated strong ability to represent complex conditional distributions and have been adapted to mitigate sparsity in sequential settings. However, these diffusion-based temporal methods (Yang et al., 2023; Liu et al., 2025a) often treat time as a static condition or operate in a discretized temporal grid rather than a continuous guiding signal for latent alignment. Meanwhile, using continuous-time ODE methods (Nguyen et al., 2022; Wang et al., 2023a; Chen et al., 2024) alone can struggle to reconstruct latent states across long unobserved intervals—exactly where system dynamics may drift the most.

In this paper, we propose Coupled Continuous-Time Latent Dynamics (CoCLD), a unified framework that jointly models individual and population-level dynamics in a shared continuous latent space. CoCLD addresses the dual challenges of data sparsity and asynchronous interaction through two core components: (i) a Diffusion-based Latent Interpolator that reconstructs continuous latent paths from sparse event anchors via a time-guided denoising process, and (ii) a Coupled Neural ODE mechanism that formalizes the mutual influence between individual and global states as a system of coupled differential equations. This architecture allows for the principled interpolation, generation, and alignment of latent states across arbitrary time points, ensuring that individual trajectories are rectified by global distributions at every infinitesimal time step.

We provide a comprehensive theoretical characterization of CoCLD, establishing the existence and uniqueness of the coupled dynamics, as well as the stability and universal expressivity of the learned trajectories. Empirical evaluations on diverse real-world datasets demonstrate that CoCLD effectively recovers complex dynamic dependencies and significantly outperforms state-of-the-art baselines. Our main contributions are summarized as follows:

- We introduce the **Co**upled **C**ontinuous-Time **L**atent **D**ynamics (**CoCLD**), a novel framework that couples individual and population-level continuous-time dynamics, bridging the gap between personalized modeling and global trend alignment.

- We synergize time-guided diffusion with Neural ODEs, enabling robust latent state reconstruction and synchronization under irregular and sparse observations. We establish formal proofs for the existence, stability, and universal approximation capabilities of the proposed coupling mechanism, providing a rigorous foundation.

- Extensive experiments across next-event prediction, trajectory generation, and sequential behavior modeling tasks validate the effectiveness and generalizability of CoCLD compared to state-of-the-art methods.

## 2. Related Work

The modeling of irregular temporal dependencies has evolved through several paradigms. The CoCLD model lies at the intersection of continuous-time dynamics, diffusion-based generative modeling, and coupled system analysis.

### 2.1. Continuous-Time Modeling and Neural ODEs

The introduction of Neural Ordinary Differential Equations (Neural ODEs) (Chen et al., 2018) has revolutionized the modeling of irregularly sampled data by defining hidden state transitions through continuous vector fields. Subsequent extensions, such as Latent ODE (Rubanova et al., 2019), Latent Time ODE (Anumasa & Srijith, 2022), and Graph ODE (Qin et al., 2024), further enabled the handling of non-uniform observations by integrating ODE solvers with recurrent architectures. Although these methods can formally model continuous trajectories, they often struggle with severe temporal sparsity (Chen et al., 2023b) and fail to capture long-range dependencies via deterministic evolution alone. By contrast, CoCLD introduces a generative interpolation mechanism to fill sparse gaps and yield more *robust latent representations*.

### 2.2. Diffusion Models for Sequential Data

Denoising Diffusion Probabilistic Models (DDPMs) (Ho et al., 2020) have set new benchmarks in generative tasks. Recently, their application has been extended to sequential recommendation (Zhao et al., 2024) and time-series forecasting (Yang et al., 2024) to alleviate data sparsity. For instance, PreferDiff (Liu et al., 2025a) introduces a diffusion-model-specific personalized ranking objective. However, most existing diffusion-based temporal models operate in discrete-time latent spaces or treat time as a static condition. In contrast, CoCLD treats the diffusion as a *Latent Interpolator* that operates over a temporal continuum, enabling the generation of consistent latent states at arbitrary time points.

### 2.3. Multi-level and Coupled Dynamics

In complex systems, individual behaviors are often intertwined with population-level distributional shifts. Previous works have attempted to capture such multi-level dependencies using hierarchical RNNs (Quadrana et al., 2017), GRU-based ODE (Guo et al., 2022), or graph-based spatial-temporal models (Fang et al., 2021; Jin et al., 2022; Linot et al., 2023). But they typically rely on discrete-time updates or late-fusion strategies, which fail to capture the asynchronous and infinitesimal interactions between individual and global states. CoCLD fills this gap by formulating the interaction through a system of coupled differential equations, providing a unified framework for aligning asynchronous processes in a *shared continuous-time latent space*.

# 3. Method

In this section, we present the **Co**upled **C**ontinuous-Time **L**atent **D**ynamics (**CoCLD**) framework. As illustrated in Figure 1, CoCLD is designed to model the continuous evolution of individual entities under irregular observations while simultaneously capturing the influence of population-level distributional shifts. The framework consists of two core components: a *Diffusion-based Latent Interpolator* that reconstructs continuous-time latent states from sparse events, and a *Coupled Neural ODE* mechanism that aligns and evolves individual and global dynamics.

## 3.1. Problem Formulation

Consider a population of entities (e.g., users, agents, vehicles), where the $i$-th entity is associated with an irregularly sampled event sequence $\mathcal{S}_i = \{(e_{i,k}, t_{i,k})\}_{k=1}^{L_i}$. Here, $e_{i,k} \in \mathbb{R}^{d_e}$ represents the observed event feature at timestamp $t_{i,k}$, and $t_{i,k} \in [0, T]$. The time intervals $\Delta t_k = t_{i,k+1} - t_{i,k}$ are non-uniform and stochastic. Simultaneously, the system is governed by a global distributional process (e.g., seasonal trends, global mobility patterns).

Our goal is to learn a continuous-time mapping that models the conditional probability of a future event $e_{i,L_i+1}$ at time $t_{target}$, given the history $\mathcal{S}_i$ and the global context $\mathcal{C}$. We hypothesize that the observed events are sparse realizations of an *individual latent state* $\mathbf{h}_{ind}^{(i)}(t)$, which evolves via a differential equation coupled with a *global latent state* $\mathbf{h}_{glob}(t)$. CoCLD aims to learn a coupling mechanism aligning the two in a shared continuous-time latent space to support downstream tasks (e.g., next-event prediction, trajectory generation, sequential behavior modeling)

## 3.2. Diffusion-based Latent Interpolator

A fundamental challenge in modeling irregular events is the temporal sparsity—observations are often absent during critical intervals where dynamics shift. To address this, we propose a *Diffusion-based Latent Interpolator* that functions as a generative model to infer potential latent states at arbitrary time points $\tau$, filling the gaps in observation.

### 3.2.1. LATENT SPACE CONSTRUCTION

We first map the discrete observed sequence $\mathcal{S}_i$ into a latent distribution using a Variational Autoencoder (VAE). For a query time $\tau$, we construct a context-aware initial latent state $\mathbf{z}_0^\tau$. To capture the temporal context, we employ a learnable time encoding function $\phi(\cdot)$:

$$\mathbf{z}_0^\tau \sim q_\phi(\mathbf{z}|\mathcal{S}_i, \tau) = \mathcal{N}(\boldsymbol{\mu}_\phi, \boldsymbol{\sigma}_\phi^2), \tag{1}$$

where $[\boldsymbol{\mu}_\phi, \boldsymbol{\sigma}_\phi] = \text{Encoder}(\mathcal{S}_i, \phi(\tau))$.

### 3.2.2. TIME-GUIDED REVERSE DIFFUSION PROCESSES

We adapt the Denoising Diffusion Probabilistic Model (DDPM) to the continuous-time domain. The forward process gradually adds Gaussian noise to $\mathbf{z}_0^\tau$ over $K$ steps, producing a sequence $\{\mathbf{z}_1^\tau, \ldots, \mathbf{z}_K^\tau\}$. The reverse process aims to reconstruct the clean latent state $\mathbf{z}_0^\tau$ from noise $\mathbf{z}_K^\tau \sim \mathcal{N}(\mathbf{0}, \mathbf{I})$, conditioned on the specific time embedding $\tau$. This time-guidance is crucial, as it forces the diffusion model to generate latent states that are consistent with the dynamics at time $\tau$, rather than a generic average.

The reverse transition is parameterized as:

$$p_\theta(\mathbf{z}_{k-1}^\tau|\mathbf{z}_k^\tau, \tau) = \mathcal{N}(\mathbf{z}_{k-1}^\tau; \boldsymbol{\mu}_\theta(\mathbf{z}_k^\tau, \tau, k), \boldsymbol{\Sigma}_\theta(\mathbf{z}_k^\tau, \tau, k)), \tag{2}$$

By sampling from this process, we obtain an interpolated latent state $\hat{\mathbf{z}}_0^\tau$ that represents the inferred individual status at time $\tau$. This effectively transforms the sparse event sequence into a dense, continuous-time latent trajectory.

To provide a statistical foundation for our diffusion-based latent reconstruction, we analyze its convergence properties with respect to the underlying data distribution. Proposition 3.1 ensures that our time-guided interpolation yields a statistically consistent estimator of the latent state, even under extreme temporal sparsity (see Appendix F.1 for full derivation).

**Proposition 3.1** (Consistency of Diffusion Interpolation)**.** *Let $p(\mathbf{z}_0^\tau|\mathcal{S}_i)$ be the true posterior distribution of the latent state. The time-guided diffusion process in CoCLD, by minimizing the Evidence Lower Bound (ELBO) in Equation (9), yields a variational distribution $q_\theta(\hat{\mathbf{z}}_0^\tau)$ that minimizes the KL-divergence $D_{KL}(q_\theta||p)$, ensuring that $\hat{\mathbf{z}}_0^\tau$ is an unbiased estimator of the latent state under sparse observations.*

## 3.3. Coupled Neural Ordinary Differential Equations

While the diffusion module handles data sparsity, it does not explicitly model the dynamic laws governing the system. We introduce a Coupled Neural ODE framework to model the evolution of individual states $\mathbf{h}_{ind}(t)$ and the global state $\mathbf{h}_{glob}(t)$ as a unified dynamical system.

### 3.3.1. INDIVIDUAL-GLOBAL COUPLING

Existing methods often treat individual sequences and global trends independently. However, in real-world systems, these processes are asynchronous but interactive. We formalize this interaction as a system of coupled differential equations:

$$\frac{d\mathbf{h}_{ind}(t)}{dt} = f_\phi(\mathbf{h}_{ind}(t), \psi(\mathbf{h}_{glob}(t)), t), \tag{3}$$

$$\frac{d\mathbf{h}_{glob}(t)}{dt} = g_\psi(\mathbf{h}_{glob}(t), \rho(\{\mathbf{h}_{ind}^{(i)}(t)\}_i), t), \tag{4}$$

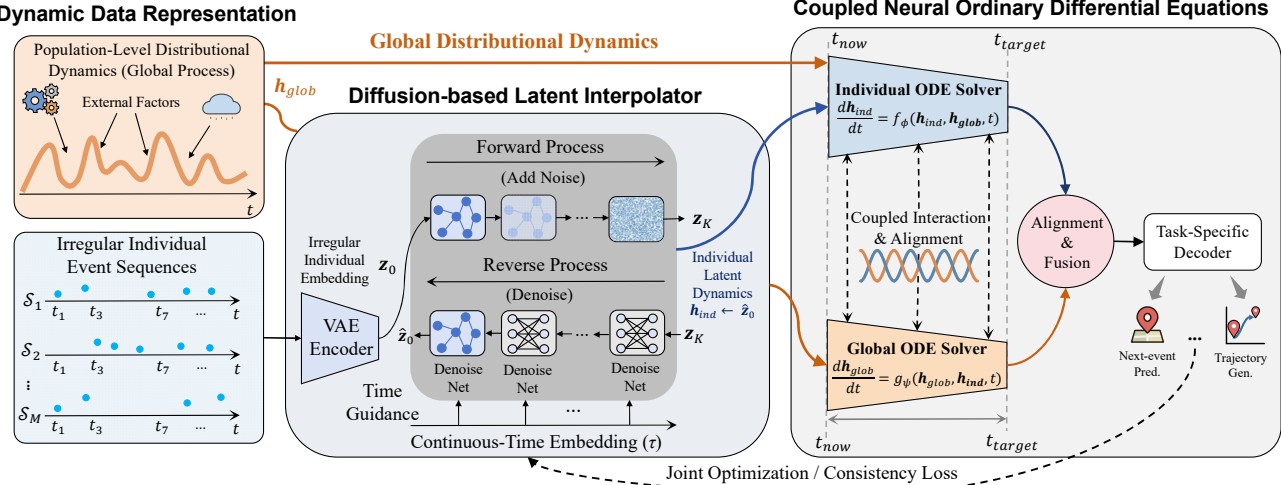

Figure 1. The overall architecture of the proposed Coupled Continuous-Time Latent Dynamics (CoCLD) framework.

where $f_\phi(\cdot)$ and $g_\psi(\cdot)$ are neural networks parameterizing the vector fields.

- Equation (3) describes the *individual dynamics*. The term $\psi(\mathbf{h}_{glob}(t))$ represents the influence of the global context on the individual (e.g., a population-level trend influencing a user's next action).

- Equation (4) describes the *global distributional dynamics*. It evolves based on its own history and potentially aggregates feedback from individuals via $\rho(\mathbf{h}_{ind}^{(i)}(t))$.

The coupled differential equations defined above formalize a complex interplay between individual and global latent processes. To ensure that this unified dynamical system is mathematically well-posed and stable during numerical integration, we establish Theorem 3.2 and full derivation based on the Picard–Lindelöf theorem in Appendix F.2.

**Theorem 3.2** (Existence and Uniqueness of Coupled Dynamics). *Assume that the vector fields $f_\phi$ and $g_\psi$ in Equation (3) and Equation (4) are Lipschitz continuous in both latent states $\mathbf{h}_{ind}, \mathbf{h}_{glob}$ and continuous in time $t$. Given initial latent states $\mathbf{h}_{ind}(0)$ and $\mathbf{h}_{glob}(0)$ reconstructed by the Diffusion Interpolator, there exists a unique solution $(\mathbf{h}_{ind}(t), \mathbf{h}_{glob}(t))$ for the coupled system over the time.*

### 3.3.2. ALIGNMENT AND FUSION

While the ODE solver enables continuous-time state evolution, the accuracy of the final prediction $\mathbf{h}(t_{target})$ is inherently tied to the precision of the initial state reconstructed by the diffusion interpolator. To characterize how reconstruction errors propagate through the coupling mechanism, we provide the following Lemma 3.3 stability bound, with more details available in Appendix F.3.

**Lemma 3.3** (Stability of Latent Trajectories). *Let $\hat{\mathbf{z}}_0^\tau$ be the interpolated state with a reconstruction error $\epsilon = \|\mathbf{z}_0^\tau - \hat{\mathbf{z}}_0^\tau\|_2$. Under the Lipschitz constant $L_f$ of the vector field $f_\phi$, the discrepancy between the predicted latent state $\hat{\mathbf{h}}_{ind}(t)$ and the true latent path $\mathbf{h}_{ind}(t)$ at $t_{target}$ is bounded by:*

$$\|\mathbf{h}_{ind}(t_{target}) - \hat{\mathbf{h}}_{ind}(t_{target})\| \le \epsilon \cdot e^{(L_f|t_{target}-\tau|)}. \quad (5)$$

To solve for the state at a target time $t_{target}$, we introduce an ODESolver (e.g., Runge-Kutta) starting from the initial states derived from the Diffusion Interpolator at $t_{now}$:

$$\mathbf{h}_{ind}(t_{target}) = \text{ODE}_{\text{Solver}}(f_\phi, \mathbf{h}_{ind}(t_{now}), t_{target}), \quad (6)$$

$$\mathbf{h}_{glob}(t_{target}) = \text{ODE}_{\text{Solver}}(g_\psi, \mathbf{h}_{glob}(t_{now}), t_{target}). \quad (7)$$

Beyond stability and existence, it is essential to verify whether CoCLD is expressive enough to capture the diverse patterns of real-world event sequences. We demonstrate the representational capacity of CoCLD by showing that it can approximate any continuous joint latent trajectory with arbitrary precision in Theorem 3.4. The complete proof is detailed in Appendix F.4.

**Theorem 3.4** (Universal Approximation of Coupled Continuous Dynamics). *For any joint continuous trajectory $(\mathcal{H}_{ind}, \mathcal{H}_{glob})$ defined on $[0, T]$, there exists a CoCLD configuration with sufficiently large hidden dimensions such that for any $\delta > 0$, the learned coupled ODE system satisfies:* $\sup_{t\in[0,T]} \|[\mathbf{h}_{ind}(t), \mathbf{h}_{glob}(t)] - [\mathcal{H}_{ind}(t), \mathcal{H}_{glob}(t)]\| < \delta$.

The alignment mechanism ensures that the individual's trajectory is rectified by the global distribution at every infinitesimal time step. Finally, a task-specific decoder maps the coupled state to the prediction:

$$\hat{y} = \text{Decoder}(\mathbf{h}_{ind}(t_{target}) \oplus \text{Attn}(\mathbf{h}_{ind}, \mathbf{h}_{glob})). \quad (8)$$

## 3.4. Optimization

The training of CoCLD involves a joint optimization objective comprising the diffusion reconstruction loss and the task-specific prediction loss. Appendix C provides a detailed training process.

**Diffusion Loss ($\mathcal{L}_{diff}$).** We optimize the ELBO for the diffusion process to ensure accurate latent state interpolation:

$$\mathcal{L}_{diff} = \mathbb{E}_{\tau \sim U(0,T)} \left[ \|\mathbf{z}_0^\tau - \hat{\mathbf{z}}_0^\tau\|^2 + \lambda_{KL} D_{KL}(q_\phi \| p_\theta) \right]. \quad (9)$$

**Prediction Loss ($\mathcal{L}_{task}$).** Depending on the specific downstream task (e.g., classification or regression).

**Total Loss.** The final objective is $\mathcal{J} = \mathcal{L}_{task} + \lambda_1 \mathcal{L}_{diff} + \lambda_2 \mathcal{L}_{KL}$. We employ an iterative training strategy where the diffusion module and the ODE module are updated alternately to stabilize the learning of coupled dynamics.

## 3.5. Disucssion and Analysis

To evaluate the efficiency of CoCLD, we analyze its time and space complexity with respect to the number of entities $M$, sequence length $L$, latent dimension $d$, diffusion steps $K$, and ODE solver evaluations $N_{ODE}$.

**Time Complexity.** The computational cost is primarily distributed across two main modules. i) Diffusion-based Latent Interpolator: For each entity, the VAE-based encoding of a sequence of length $L$ requires $\mathcal{O}(L \times d^2)$. During the reverse diffusion process, the model performs $K$ denoising steps. Each step involves a forward pass through the denoising network, costing $\mathcal{O}(d^2)$. Thus, the total complexity for interpolation across the entire population is $\mathcal{O}(M \times (L + K) \times d^2)$. ii) Coupled Neural ODEs: In each step of the ODE solver (e.g., Runge-Kutta), the vector fields $f_\phi$ and $g_\psi$ are evaluated. $f_\phi$ computes individual dynamics for $M$ entities with $\mathcal{O}(M \times d^2)$. $g_\psi$ updates the global state by aggregating individual representations via $\rho(\cdot)$, which takes $\mathcal{O}(M \times d)$ for operations like mean-pooling, plus $\mathcal{O}(d^2)$ for the global transition. For $N_{ODE}$ solver steps, the complexity is $\mathcal{O}(N_{ODE} \times M \times d^2)$. Overall, the time complexity is $\mathcal{O}(M \times (L + K + N_{ODE}) \times d^2)$, which scales linearly with the number of entities $M$, making it suitable for large-scale systems.

**Space Complexity.** The memory footprint is dominated by the storage of latent trajectories and model parameters. i) Parameters: The networks $f_\phi$, $g_\psi$, and the diffusion denoising net utilize $O(d^2)$ parameters. ii) Memory: Storing the individual latent states and the diffusion chain requires $O(M \times (L + K) \times d)$. If the adjoint sensitivity method is employed during training, the memory cost for the ODE module remains constant relative to $N_{ODE}$, resulting in an overall space complexity of $O(M \times (L + K) \times d)$.

## 4. Experiments

To validate the effectiveness of CoCLD, we conduct extensive evaluations on three distinct tasks: (1) Next-event prediction (sparse event forecasting), (2) Continuous trajectory generation, and (3) Sequential behavior modeling.

## 4.1. Experimental Setup

**Datasets.** We utilize five large-scale benchmark datasets for Next-event Prediction: (a) IST, (b) NYC, (c) DC, (d) Gowalla[1], and (e) Brightkite[2], with the first three derived from Foursquare[3]. These datasets are characterized by high sparsity and irregular time intervals. For Trajectory Generation, we employ two mobility datasets: (a) Chengdu and (b) Xi'an, which contain continuous GPS trajectories from taxi operations Didi[4]. For Sequential Behavior Modeling, we use three interaction records between users and items in different fields on the Amazon platform[5]: (a) Beauty, (b) Sports, and (c) Toys. More details are in Appendix A.

**Baselines.** We compare CoCLD against a diverse set of state-of-the-art methods: (1) Discrete-time Models: SAS-Rec (Kang & McAuley, 2018) and LightGCN (He et al., 2020). (2) ODE-based Continuous-time Models: GNG-ODE (Guo et al., 2022) and SGODE (Chen et al., 2024). (3) Diffusion-based Generative Models: DiffRec (Wang et al., 2023b), DreamRec (Yang et al., 2023), DDRM (Zhao et al., 2024), and PreferDiff (Liu et al., 2025a). (4) Trajectory Generators: VAE (Xia et al., 2018), TrajGAN (Liu et al., 2018), DP-TrajGAN (Zhang et al., 2023), Diffwave (Kong et al., 2020), and DiffTraj (Zhu et al., 2023).

**Implementation Details.** We implemented CoCLD with PyTorch 3.8 and conducted it on a NVIDIA A800 80GB GPU. Following the joint objective described in Section 3.4, we update the diffusion interpolator and the coupled ODE component in an interleaved manner to stabilize training. We use Adam with learning rate $1e-3$ and weight decay $1e-3$, batch size 128, and train up to 100 epochs with early stopping (patience 5). For the diffusion interpolator, we use $K = 5$ denoising steps with a linear noise schedule; during inference, we set the forward diffusion steps to 0. For continuous-time evolution, we solve the coupled dynamics with a Runge–Kutta (RK4) ODE solver with step size $\Delta t = 0.2$. Additional hyperparameter configurations and task-specific settings are provided in Appendix E.

---

[1]http://snap.stanford.edu/data/loc-Gowalla.html

[2]http://snap.stanford.edu/data/loc-Brightkite.html

[3]https://sites.google.com/site/yangdingqi/home/foursquare-dataset

[4]https://outreach.didichuxing.com/

[5]https://cseweb.ucsd.edu/ jmcauley/datasets.html

*Table 1.* Performance comparison on next-event prediction. The best results are highlighted in **bold**, and the second best are underlined.

| Dataset | Metrics | SASRec | LightGCN | GNG-ODE | SGODE | DiffRec | DreamRec | DDRM | PreferDiff | **CoCLD** |
|---------|---------|--------|----------|---------|-------|---------|----------|------|------------|-----------|
| IST | Acc@5 | 0.2141 | 0.2417 | 0.2400 | 0.2714 | 0.2854 | 0.2887 | 0.3125 | 0.3285 | **0.3486** |
| | Acc@10 | 0.2796 | 0.2824 | 0.2899 | 0.3605 | 0.3604 | 0.3530 | 0.3830 | 0.3981 | **0.4255** |
| | NDCG@5 | 0.1774 | 0.1911 | 0.1922 | 0.2356 | 0.2301 | 0.2314 | 0.2504 | 0.2587 | **0.2706** |
| | NDCG@10 | 0.1848 | 0.2044 | 0.2088 | 0.2557 | 0.2544 | 0.2528 | 0.2739 | 0.2817 | **0.2964** |
| NYC | Acc@5 | 0.2443 | 0.3016 | 0.3302 | 0.3645 | 0.3533 | 0.3196 | 0.3205 | 0.3838 | **0.3937** |
| | Acc@10 | 0.2907 | 0.3620 | 0.4023 | 0.4334 | 0.4225 | 0.4020 | 0.4088 | 0.4755 | **0.5083** |
| | NDCG@5 | 0.1935 | 0.2011 | 0.2232 | 0.2762 | 0.2419 | 0.2417 | 0.2691 | 0.2705 | **0.2801** |
| | NDCG@10 | 0.2163 | 0.2364 | 0.2571 | 0.3024 | 0.2846 | 0.2840 | 0.2978 | 0.3008 | **0.3179** |
| DC | Acc@5 | 0.2048 | 0.2024 | 0.2680 | 0.2517 | 0.2843 | 0.2896 | 0.2901 | 0.2963 | **0.3090** |
| | Acc@10 | 0.2963 | 0.3090 | 0.3210 | 0.3037 | 0.3448 | 0.3420 | 0.3685 | 0.3655 | **0.3827** |
| | NDCG@5 | 0.1869 | 0.1859 | 0.2131 | 0.1995 | 0.2226 | 0.2250 | 0.2307 | 0.2309 | **0.2368** |
| | NDCG@10 | 0.2109 | 0.2168 | 0.2356 | 0.2169 | 0.2439 | 0.2492 | 0.2540 | 0.2545 | **0.2620** |
| Gowalla | Acc@5 | 0.1675 | 0.1943 | 0.2019 | 0.2354 | 0.2604 | 0.2806 | 0.2764 | 0.2857 | **0.3003** |
| | Acc@10 | 0.2578 | 0.2737 | 0.2758 | 0.3001 | 0.3385 | 0.3710 | 0.3632 | 0.3700 | **0.3775** |
| | NDCG@5 | 0.2880 | 0.3587 | 0.3585 | 0.3646 | 0.3812 | 0.4215 | 0.4159 | 0.4221 | **0.4313** |
| | NDCG@10 | 0.3271 | 0.3664 | 0.3622 | 0.3860 | 0.4254 | 0.4659 | 0.4627 | 0.4688 | **0.4725** |
| Brightkite | Acc@5 | 0.1738 | 0.2448 | 0.2721 | 0.2878 | 0.2943 | 0.2753 | 0.2989 | 0.3058 | **0.3158** |
| | Acc@10 | 0.2220 | 0.2918 | 0.3216 | 0.3335 | 0.3404 | 0.3248 | 0.3478 | 0.3584 | **0.3649** |
| | NDCG@5 | 0.1630 | 0.2328 | 0.2602 | 0.2750 | 0.2824 | 0.2635 | 0.2863 | 0.2935 | **0.3025** |
| | NDCG@10 | 0.1847 | 0.2539 | 0.2825 | 0.2955 | 0.3030 | 0.2857 | 0.3083 | 0.3171 | **0.3245** |

## 4.2. Task 1: Next-Event Prediction

In this task, we aim to predict the next event (e.g., location visit) given a historical sequence. This assesses the model's ability to capture temporal dependencies in sparse data. We use Acc@k and NDCG@k as evaluation metrics (k=5, 10).

The results in Table 2 demonstrate that CoCLD consistently outperforms all baselines across datasets.

- **Comparison with Discrete-time Models:** Methods like SASRec suffer in performance due to the loss of fine-grained temporal information caused by discrete time-steps. CoCLD's continuous modeling captures the precise timing of events.

- **Comparison with Diffusion-based Baselines:** While DiffRec and PreferDiff leverage generative capabilities, they lack explicit dynamic modeling. CoCLD integrates diffusion as an interpolator within a dynamical system, providing better guidance for state evolution.

- **Comparison with ODE Continuous-time Models:** Although SGODE models continuous dynamics, it treats entities in isolation or uses static graph structures. CoCLD's superiority stems from the *Coupled Neural ODE* mechanism, which dynamically aligns individual trajectories with population-level distributional shifts.

## 4.3. Task 2: Trajectory Generation

In this task, we evaluate the quality of generated mobility trajectories. We measure the distributional similarity between

*Table 2.* Performance comparison on trajectory generation. The best results are highlighted in **bold**. Lower ($\downarrow$) is better.

| Methods | Chengdu | | Xi'an | |
|---------|---------|------|-------|------|
| | Density ($\downarrow$) | Trip ($\downarrow$) | Density ($\downarrow$) | Trip ($\downarrow$) |
| VAE | 0.0142 | 0.0515 | 0.0239 | 0.0612 |
| TrajGAN | 0.0133 | 0.0476 | 0.0218 | 0.0510 |
| DP-TrajGAN | 0.0130 | 0.0442 | 0.0207 | 0.0498 |
| Diffwave | 0.0143 | 0.0251 | 0.0211 | 0.0338 |
| DiffTraj | 0.0059 | 0.0156 | 0.0128 | 0.0171 |
| **CoCLD** | **0.0041** | **0.0122** | **0.0123** | **0.0135** |

generated and real trajectories using Jensen-Shannon Divergence (JSD) on density error and trip error. See Appendix D for formal definitions.

As shown in Table 2, CoCLD achieves the lowest JSD scores in both density error and trip error. This indicates that our coupled dynamics not only learn individual motion patterns but also preserve the global spatial-temporal distribution of the crowd, which standard GANs or single-agent diffusion models (e.g., Diffwave and DiffTraj) often fail to capture.

To qualitatively assess the generation fidelity, we visualize the generated trajectories against real ground-truth paths on the Chengdu dataset in Figure 2. CoCLD generates trajectories that are not only locally smooth but also globally consistent with the city's underlying flow dynamics. The coupled ODE mechanism ensures that the generated agents follow realistic population-level trends, effectively preserving the semantic structure of the urban mobility network.

*Table 3.* Performance comparison on sequential behavior modeling. Best results are **bolded**, second best are underlined.

| Model | Amazon-Beauty | | | Amazon-Sports | | | Amazon-Toys | | |
|---|---|---|---|---|---|---|---|---|---|
| | R@5 | R@10 | MRR | R@5 | R@10 | MRR | R@5 | R@10 | MRR |
| GRU4Rec | 0.0133 | 0.0225 | 0.0085 | 0.0115 | 0.0167 | 0.0078 | 0.0077 | 0.0121 | 0.0049 |
| BERT4Rec | 0.0159 | 0.0263 | 0.0091 | 0.0109 | 0.0160 | 0.0065 | 0.0127 | 0.0186 | 0.0074 |
| SASRec | 0.0225 | 0.0360 | 0.0131 | 0.0112 | 0.0168 | 0.0060 | 0.0172 | 0.0244 | 0.0099 |
| GNG-ODE | 0.0329 | 0.0549 | 0.0205 | 0.0192 | 0.0325 | 0.0120 | 0.0184 | 0.0287 | 0.0118 |
| SGODE | 0.0260 | 0.0451 | 0.0162 | 0.0154 | 0.0277 | 0.0105 | 0.0189 | 0.0293 | 0.0129 |
| DiffRec | 0.0115 | 0.0190 | 0.0089 | 0.0119 | 0.0171 | 0.0085 | 0.0066 | 0.0117 | 0.0049 |
| DreamRec | 0.0135 | 0.0193 | 0.0096 | 0.0128 | 0.0174 | 0.0088 | 0.0091 | 0.0127 | 0.0080 |
| DDRM | 0.0221 | 0.0344 | 0.0133 | 0.0157 | 0.0222 | 0.0103 | 0.0085 | 0.0192 | 0.0071 |
| PreferDiff | 0.0286 | 0.0573 | 0.0126 | 0.0171 | 0.0340 | 0.0138 | 0.0305 | 0.0509 | 0.0185 |
| **CoCLD** | **0.0434** | **0.0715** | **0.0249** | **0.0264** | **0.0431** | **0.0165** | **0.0402** | **0.0652** | **0.0238** |

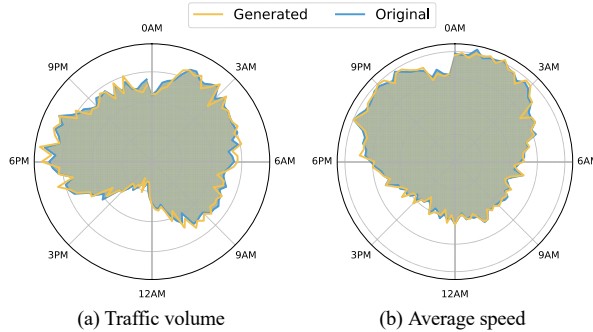

(a) Traffic volume      (b) Average speed

*Figure 2.* Comparison between generated trajectories and original.

### 4.4. Task 3: Sequential Behavior Modeling

To further evaluate CoCLD's capability in capturing complex user behavioral patterns over time, we conduct experiments on three sub-categories of the Amazon Review dataset. These datasets record user interactions with timestamps, characterized by highly variable intervals between behaviors. Beyond the existing ODE and diffusion-based models, we incorporate two sequence modeling baselines based on RNN/Transformer architectures: GRU4Rec (Hidasi et al., 2015) and BERT4Rec (Sun et al., 2019).

We report Recall@k (R@k) and Mean Reciprocal Rank (MRR) in Table 3. CoCLD achieves state-of-the-art performance across all metrics. Notably, in the *Sports* dataset where seasonal trends (global dynamics) heavily influence individual purchases, CoCLD outperforms the strongest baseline (*PreferDiff*) by a significant margin (e.g., $+19.57\%$ in MRR). PreferDiff treats sequence modeling as a direct conditional generation task, which can suffer from mode collapse when supervision signals are sparse. CoCLD adopts a "Interpolate-then-Evolve" paradigm. This validates that explicitly coupling individual latent states with population-level evolution allows the model to better anticipate behavior shifts driven by external temporal factors.

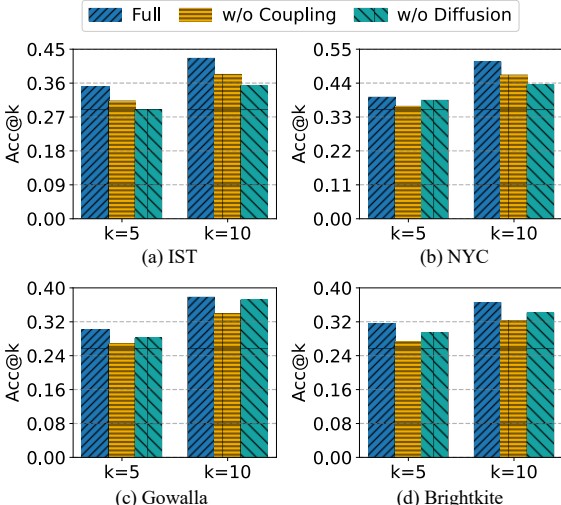

*Figure 3.* Results of the ablation study.

### 4.5. Ablation Study

To provide deeper insights into the mechanisms of CoCLD, we conduct component ablation. We create three variants: (a) *Full Model*. (b) *w/o Coupling*: Replaces the coupled ODE system with independent ODEs for each entity, removing the global guidance $g_\psi$. (c) *w/o Diffusion*: Replacing the diffusion interpolator with a standard linear interpolation. The results in Figure 3 show that removing the Coupling mechanism leads to an average $21.5\%$ drop in accuracy, confirming that individual behavior is significantly modulated by global trends. Removing Diffusion leads to an average $17.6\%$ drop, highlighting the importance of high-quality latent interpolation for irregular sequences. Furthermore, the obvious degradation in *w/o Diffusion* on the IST dataset highlights the critical role of generative interpolation in bridging sparse observational gaps. Appendix G provides additional experimental results.

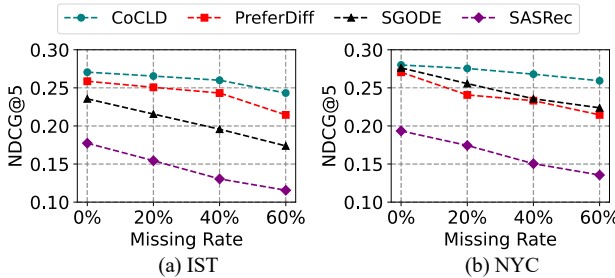

Figure 4. Performance degradation under increasing data sparsity ratios (Drop rates from 20% to 60%).

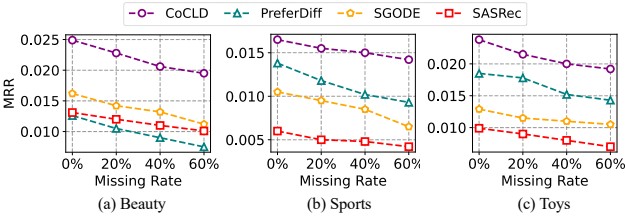

Figure 5. Robustness under increasing data sparsity ratios.

## 4.6. Robustness to Irregularity and Sparsity

Figure 4 illustrates the model performance as we randomly drop observed events (increasing sparsity). While discrete-time baselines (e.g., SASRec) degrade rapidly, CoCLD maintains robust performance. This validates our theoretical claim in Proposition 3.1 that the diffusion interpolator effectively bridges sparse intervals, preventing the "loss of tracking" common in purely continuous models (e.g., SGODE). Another key theoretical advantage of CoCLD is its stability under sparse observations (Lemma 3.3). To verify this empirically, we randomly drop observed events in the Amazon Review dataset. Figure 5 plots the performance decay in terms of MRR. We observe that CoCLD maintains relatively high performance even with 60% data missing. This empirical evidences support our claim that the diffusion-based interpolator effectively acts as a "soft bridge," reconstructing faithful latent trajectories where observations are absent.

## 4.7. Long-term Dynamics and Stability

To assess whether CoCLD captures true underlying dynamics rather than short-term correlations, we evaluate its performance on Multi-step Prediction (predicting the next $s$ events recursively) in Figure 6. In experiments on the Gowalla and Brightkite datasets, as the prediction horizon $s$ increases from 1 to 10, the performance of the Discrete-time-based model (SADRec) degrades by over 20%. CoCLD limits the degradation to roughly 10%. This stability attests to the effectiveness of the *Coupled Neural ODE*, which ensures that the extrapolated latent trajectory remains consistent with the global distributional evolution (Theorem 3.2), preventing the "drift" often seen in unconstrained latent dynamic

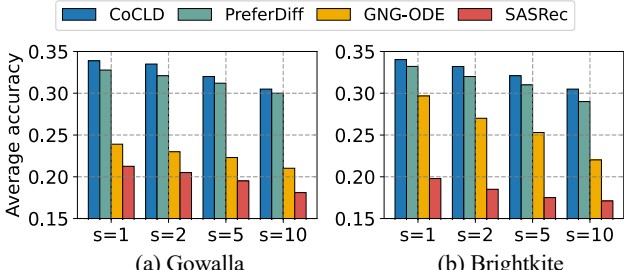

Figure 6. Performance of long-term dynamics.

Table 4. Performance comparison (Acc@10) with NPPs.

| Dataset | NHP | FNPP | THP | Auto-NPP | CoCLD |
|---|---|---|---|---|---|
| IST | 0.3568 | 0.2735 | 0.2950 | 0.3812 | **0.4255** |
| NYC | 0.3972 | 0.2935 | 0.4113 | 0.4357 | **0.5083** |
| DC | 0.3359 | 0.3051 | 0.3307 | 0.3437 | **0.3827** |
| Gowalla | 0.3597 | 0.2750 | 0.2840 | 0.3511 | **0.3775** |
| Brightkite | 0.3051 | 0.2906 | 0.3310 | 0.3413 | **0.3649** |

models. In other words, by treating the latent state evolution as a continuous flow governed by a well-posed differential equation, CoCLD ensures that the extrapolated trajectories remain strictly bounded and consistent with the global manifold, preventing the "divergence" often observed in unconstrained recurrent models.

## 4.8. Comparison with Neural Point Process

We conducted additional experiments on five datasets using the neural point process models (NPPs), including Neural Hawkes Process (NHP) (Mei & Eisner, 2017), Fully Neural Point Process (FNPP) (Omi et al., 2019), Transformer Hawkes Process (THP) (Zuo et al., 2020), and Automatic Integration for Fast and Interpretable Neural Point Processes (Auto-NPP) (Zhou & Yu, 2023). As shown in Table 4, while the NPPs are elegant and powerful for modeling the conditional intensity of irregular events, CoCLD's superiority stems from addressing two specific challenges that standard NPPs do not fully resolve:

For the coupled individual-global dynamics: NPPs primarily focus on modeling the historical dependency of a single sequence's intensity function. However, real-world events are jointly driven by individual preferences and population-level shifts. CoCLD explicitly formalizes this mutual influence as a system of Coupled Neural ODEs (Equation (3) & Equation (4)), ensuring that individual trajectories are continuously rectified by global distributions.

For the robustness to extreme temporal sparsity: Although FNPP and Auto-NPP solve the integration bottleneck to fit exact intensity functions, they are fundamentally forward-predictive models. When observations are extremely sparse (long missing intervals), their hidden states can suffer from

"drift". In contrast, CoCLD introduces a Diffusion-based Latent Interpolator that reconstructs continuous latent paths probabilistically. Our *"Interpolate-then-Evolve"* paradigm acts as a soft bridge over unobserved intervals, providing a more robust initial state for the continuous-time evolution.

## 5. Conclusion

In this paper, we presented CoCLD, a novel framework for learning coupled continuous-time latent dynamics from irregular and sparse event sequences. By bridging the gap between individual-level trajectory modeling and population-level distributional evolution, CoCLD addresses two fundamental challenges: the asynchrony of interactions and the sparsity of observations. We introduced a Time-Guided Diffusion Interpolator to robustly reconstruct continuous latent paths and formulated a Coupled Neural ODE system to align individual states with global dynamics. Theoretical analysis confirmed the existence, uniqueness, and universal approximation capabilities of our coupling mechanism. Extensive experiments across next-event prediction, trajectory generation, and sequential behavior modeling demonstrated that CoCLD consistently outperforms state-of-the-art baselines, particularly in scenarios with high sparsity and long-range dependencies. Future work will explore extending CoCLD to causal inference frameworks to disentangle the causal effects of global interventions on individual behaviors.

## Acknowledgements

This work was partly supported by the National Key Research and Development Program of China under Grant 2023YFB3002201, the National Natural Science Foundation of China under Grant 72342026, and Fundamental Research Funds for the Central Universities under Grant 2024-6-ZD-02. Dr. Yang Zhang of University of North Texas received no financial support for this work from the above grants or any other external projects. His contribution was made independently as part of his academic research. The authors also sincerely acknowledge the valuable collaboration and insightful discussions contributed by colleagues from the participating universities.

## Impact Statement

This paper presents a method for modeling irregular event sequences using coupled continuous-time dynamics. The primary goal of this research is to advance the fundamental understanding of temporal dependency learning in machine learning. The proposed CoCLD framework has potential positive impacts in various domains, such as urban planning (optimizing traffic flow based on mobility trajectories), healthcare (monitoring patient vitals with irregular measurements), and personalized recommendation systems. By accurately capturing global trends and individual behaviors, our model can help improve the efficiency of resource allocation and service delivery.

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

# A. Dataset Description and Statistics

For *Next-event Prediction*, we conducted experiments on five location-based services and mobile applications: Foursquare, Gowalla, and Brightkite. The Foursquare includes data from three major cities: Istanbul (IST), New York City (NYC), and Washington D.C. (DC). The statistics are summarized in Table 5. For *Trajectory Generation*, we employ two GPS-mobility datasets: Chengdu and Xi'an. The comprehensive summary is concluded in Table 6. For *Sequential Behavior Modeling*, we use three interaction records between users and items in different fields on the Amazon platform. The detailed information of the three datasets is shown in Table 7.

*Table 5.* Basic dataset statistics of Foursquare, Gowalla, Brightkite (# denotes the number of).

| Dataset | IST | NYC | DC | Gowalla | Brightkite |
|---|---|---|---|---|---|
| #Entities | 9,208 | 1,083 | 1,360 | 82,039 | 29,979 |
| #Events | 11,871 | 9,989 | 3,388 | 249,493 | 85,298 |
| #Interactions | 529,067 | 179,468 | 68,407 | 4,353,556 | 3,588,383 |
| Avg.visit | 57.46 | 165.71 | 50.30 | 53.07 | 119.70 |
| Density | 0.04840 | 0.01658 | 0.01484 | 0.00021 | 0.00140 |

*Table 6.* Basic dataset statistics of Chengdu and Xi'an (# denotes the number of).

| Dataset | Chengdu | Xi'an |
|---|---|---|
| #Trajetories | 5,773,525 | 3,044,828 |
| Avg.length | 176 | 244 |
| Avg.distance | 7.42km | 5.73km |

*Table 7.* Basic dataset statistics of Amazon (# denotes the number of).

| Dataset | Beauty | Sports | Toys |
|---|---|---|---|
| #Entities | 22,363 | 35,598 | 19,412 |
| #Items | 12,101 | 18,357 | 11,924 |
| #Interactions | 198,502 | 296,337 | 167,597 |
| Avg.length | 8.87 | 8.32 | 8.63 |
| Density | 0.00073 | 0.00045 | 0.00072 |

# B. Empirical Characterization of Temporal Irregularity and Sparsity

In this section, we provide a deep dive into the statistical properties of the benchmark datasets (Foursquare, Gowalla, Brightkite, and Amazon). The following analyses aim to empirically justify the necessity of CoCLD's design—specifically, why continuous-time modeling (ODE) and generative interpolation (Diffusion) are required to capture the underlying dynamics.

Figure 7 shows the 24h-population-level distributional dynamics (global process) for Foursquare (i.e., IST, NYC, DC), Gowalla, and Brightkite.

Figure 8 shows a relative distribution within each dataset (15-min granularity).

Figure 9 is the histogram and boxplot of trajectory similarity.

Figure 10 is the exploratory data analysis to reveal statistical properties of asynchronous and coupled event sequences on the Amazon-beauty dataset.

Table 8 shows the temporal dimension statistical indicators for the Amazon-beauty dataset. We can conclude that for this dataset, there is significant temporal sparsity and behavioral irregularity. Therefore, a fundamental challenge in modeling irregular events is temporal sparsity - in dynamically changing critical intervals, observations are often missing, and effective interpolation methods are needed to fill the gaps in observations.

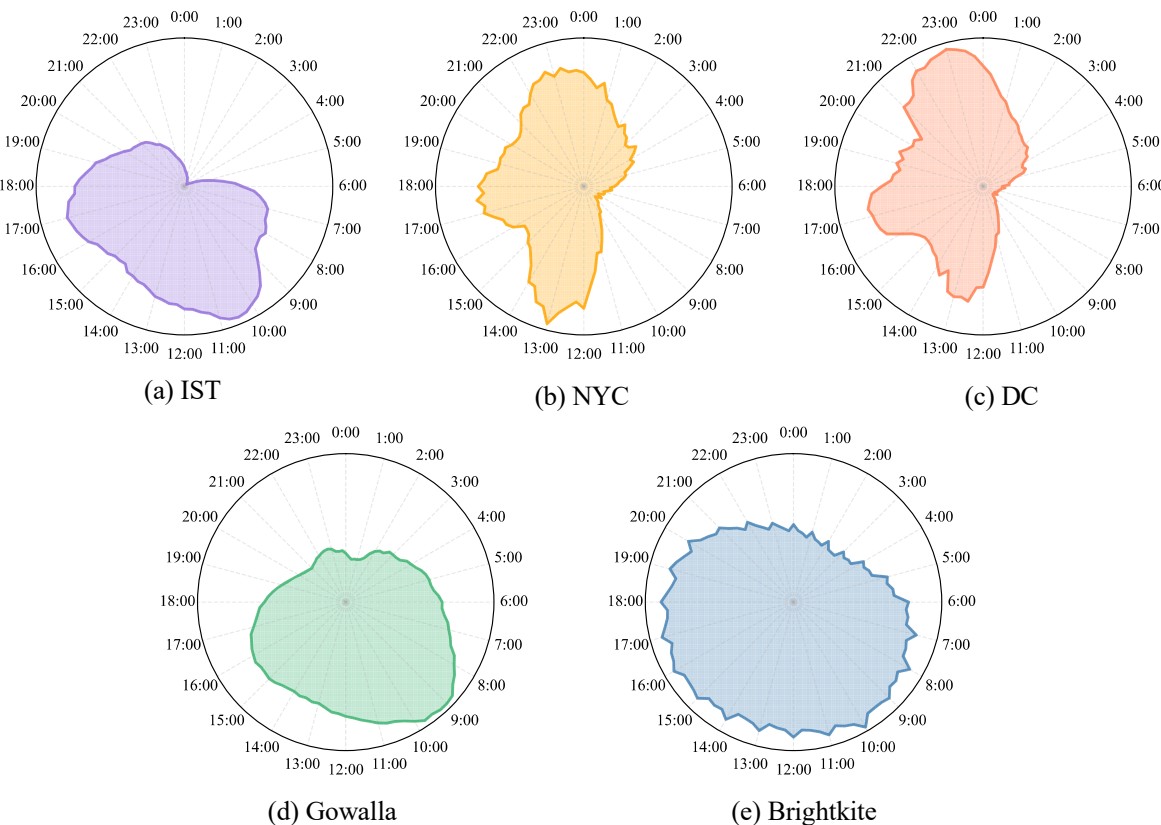

*Figure 7.* The 24h-population-level distributional dynamics (global process) for Foursquare (i.e., IST, NYC, DC), Gowalla, and Brightkite. Covered {529067, 179468, 68407, 4353556, 3588383} interactions on {IST, NYC, DC, Gowalla, Brightkite} in total.

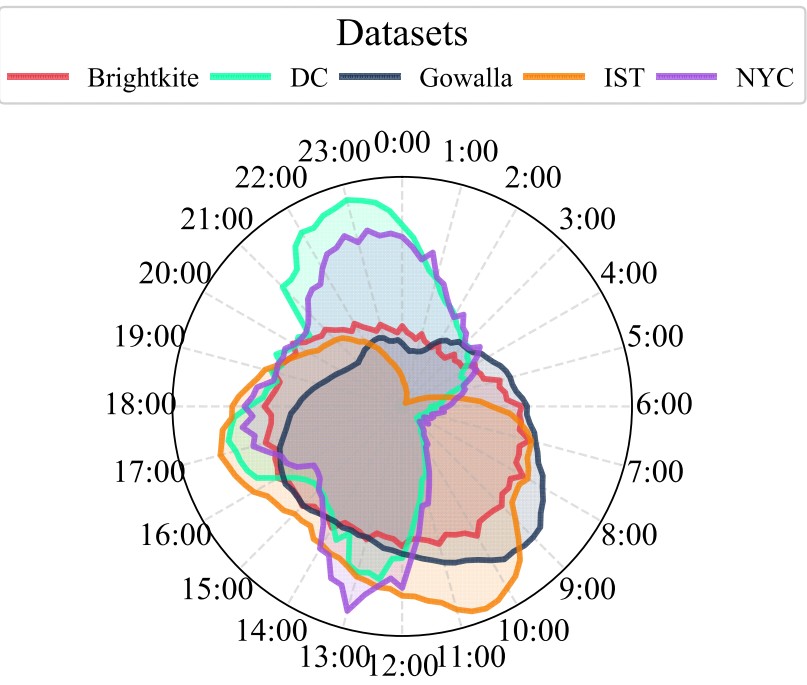

*Figure 8.* The 24h-activity pattern overlay of 5 datasets (normalized) showing relative distribution within each dataset. (a) Brightkite: peak 1.38% at 10:00; (b) DC: peak 1.97% at 23:00; (c) Gowalla: peak 1.71% at 09:15; (d) IST: peak 2.02% at 10:30; (e) NYC: peak 1.95% at 13:00.

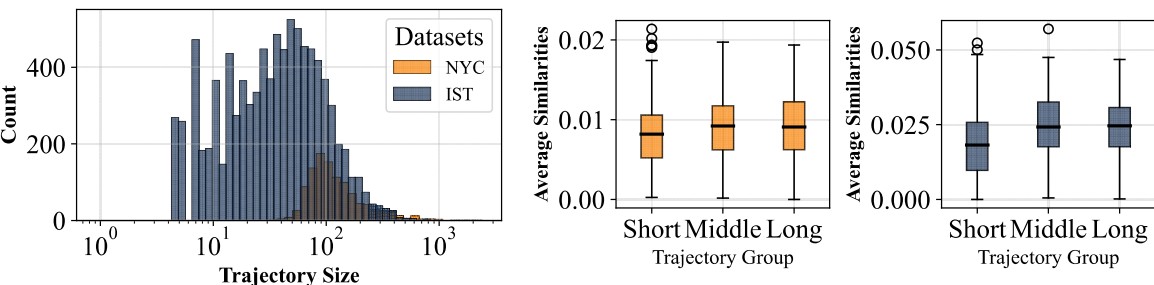

*Figure 9.* Histogram and boxplot of trajectory similarity (based on Jaccard similarity), divided into Short (bottom 30%), Middle (middle 40%), and Long (top 30%) according to trajectory length.

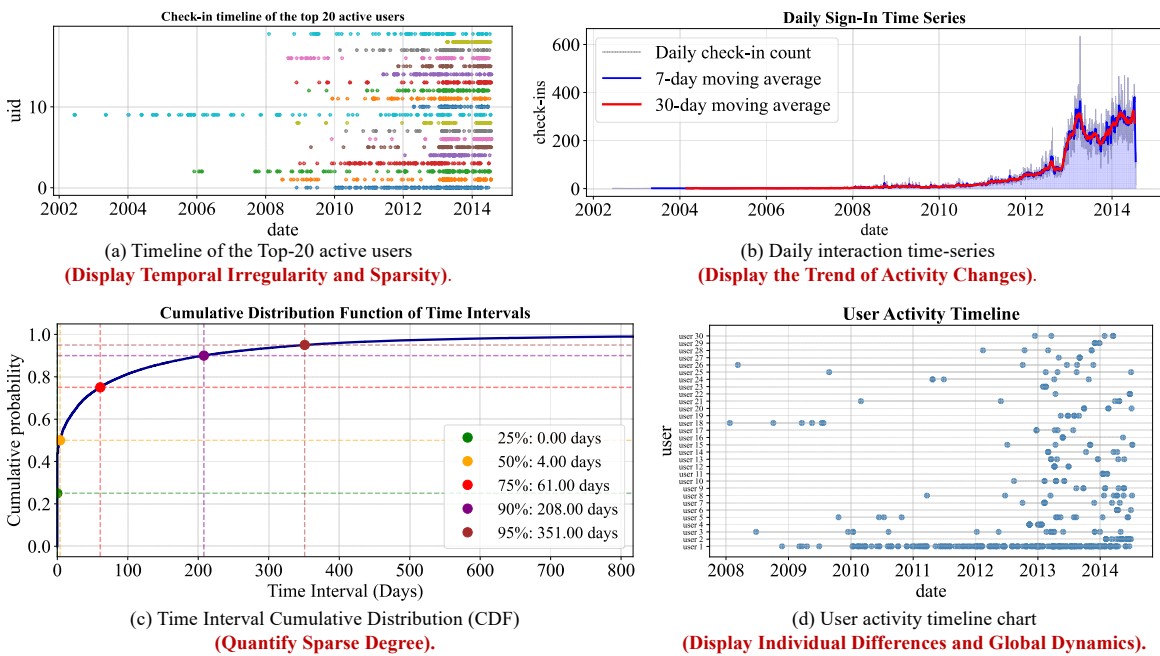

*Figure 10.* Exploratory data analysis to reveal statistical properties of asynchronous and coupled event sequences on the Amazon-beauty.

*Table 8.* Temporal statistics (time-interval based) for the Amazon-beauty dataset.

| Category / Metric | Hours | Days / % |
|---|---|---|
| **Sparsity metrics** | | |
| Mean inter-event interval | 1671.58 | 69.65 |
| Median inter-event interval | 96.00 | 4.00 |
| Total time span | – | 4424 |
| Active days | – | 2927 |
| Temporal coverage rate | – | 66.16% |
| Long-interval ratio ($> 7$ days) | – | 45.82% |
| **Irregularity metrics** | | |
| Standard deviation | 4019.28 | – |
| Coefficient of variation (CV) | 2.4045 | – |
| Burstiness coefficient ($B$) | 0.4125 | – |
| **Quantile analysis (inter-event interval)** | | |
| 25% of intervals are shorter than | 0.00 | 0.00 |
| 50% of intervals are shorter than | 96.00 | 4.00 |
| 75% of intervals are shorter than | 1464.00 | 61.00 |
| 90% of intervals are shorter than | 4992.00 | 208.00 |
| 95% of intervals are shorter than | 8424.00 | 351.00 |
| 99% of intervals are shorter than | 19608.00 | 817.00 |

*Note:* $B \approx 1$ indicates strong burstiness (events concentrated); $B \approx -1$ indicates strong regularity; $B \approx 0$ is a random Poisson-like process.

## C. Detailed Training Algorithm (PseudoCode)

Algorithm 1 formally outlines the training procedure of CoCLD from a variational inference perspective. The optimization objective is derived from the Evidence Lower Bound (ELBO) of the observed irregular sequences, which is maximized through a unified end-to-end framework.

The process consists of three theoretically distinct yet computationally integrated phases:

**Phase 1: Time-Guided Variational Interpolation.** To handle temporal sparsity, we treat the latent state at an arbitrary query time $\tau$ as a random variable. Steps 4-9 implement a **Conditional Diffusion Process**. Unlike standard diffusion, which operates on static data, our diffusion kernel $q(\mathbf{z}_0^\tau|\mathcal{S})$ is conditioned on the time-guided encoding of sparse anchors. The minimization of the reparameterized diffusion loss $\mathcal{L}_{diff}$ (Step 9) is mathematically equivalent to optimizing the variational lower bound of the latent state distribution, ensuring that the reconstructed $\hat{\mathbf{z}}_0^\tau$ serves as a statistically consistent initialization for the subsequent dynamics.

---

**Algorithm 1** Variational Learning of Coupled Continuous-Time Latent Dynamics (CoCLD)

---

1: **Input:** Irregular event sequences $\mathcal{S} = \{(t_j, v_j)\}_{i=1}^M$, global context $\mathcal{C}$, time horizon $[0, T]$.
2: **Hyperparameters:** Diffusion steps $K$, noise schedule $\{\beta_k\}_{k=1}^K$, loss weights $\lambda_1, \lambda_2$.
3: **Initialize:** Variational Encoder $\phi_{enc}$, Diffusion Net $\epsilon_\theta$, Coupled ODE Vector Fields $f_\phi, g_\psi$, Prediction Head $\omega$.
4: **while** not converged **do**
5:     Sample mini-batch of sequences $\mathcal{B} \sim \mathcal{S}$.
6:     Sample query time $\tau \sim \mathcal{U}(0, T)$ for latent interpolation.
7:     *// Phase 1: Time-Guided Variational Inference (Diffusion)*
8:     **for** each sequence $s_i \in \mathcal{B}$ **do**
9:         Encode observed anchors to distribution parameters: $\mu_i, \sigma_i = \text{Enc}_{\phi_{enc}}(s_i, \tau)$.
10:        Sample latent prior: $\mathbf{z}_0 \sim \mathcal{N}(\mu_i, \text{diag}(\sigma_i^2))$ (Reparameterization).
11:       Sample diffusion step $k \sim \mathcal{U}(1, K)$ and noise $\epsilon \sim \mathcal{N}(\mathbf{0}, \mathbf{I})$.
12:       Forward Diffusion (Corruption): $\mathbf{z}_k = \sqrt{\bar{\alpha}_k}\mathbf{z}_0 + \sqrt{1 - \bar{\alpha}_k}\epsilon$.
13:       Compute ELBO-approximated Diffusion Loss:
14:         $\mathcal{L}_{diff}^{(i)} = \|\epsilon - \epsilon_\theta(\mathbf{z}_k, k, \tau, \mathcal{C})\|^2$.
15:       Reconstruct interpolated state: $\hat{\mathbf{z}}_0^\tau \leftarrow \text{Denoise}(\mathbf{z}_K, \theta)$.
16:     **end for**
17:     *// Phase 2: Coupled Continuous-Time Evolution*
18:     Define Joint State $\mathbf{H}(t) = [\mathbf{H}_{ind}(t); \mathbf{h}_{glob}(t)]$.
19:     Set Initial Condition: $\mathbf{H}(\tau) \leftarrow [\{\hat{\mathbf{z}}_0^\tau\}_{i\in\mathcal{B}}; \mathbf{h}_{glob}(\tau)]$.
20:     Define Coupled Vector Field:
21:         $\frac{d\mathbf{H}(t)}{dt} = [\{f_\phi(\mathbf{h}_i(t), \mathbf{h}_{glob}(t))\}_{i\in\mathcal{B}}; g_\psi(\mathbf{h}_{glob}(t), \text{Agg}(\{\mathbf{h}_i(t)\}))]$.
22:     Integrate via ODESolver (using Adjoint Method for $\mathcal{O}(1)$ memory):
23:         $\mathbf{H}(t_{target}) = \mathbf{H}(\tau) + \int_\tau^{t_{target}} \frac{d\mathbf{H}(t)}{dt} dt$.
24:     *// Phase 3: Optimization via Gradient Descent*
25:     Compute Task Loss (Negative Log-Likelihood):
26:         $\mathcal{L}_{task} = -\sum_{i\in\mathcal{B}} \log p_\omega(y_i|\mathbf{h}_i(t_{target}))$.
27:     Total Objective (Negative ELBO): $\mathcal{J} = \mathcal{L}_{task} + \lambda_1\mathcal{L}_{diff} + \lambda_2\mathcal{L}_{KL}(\text{Prior})$.
28:     Update parameters $\Theta = \{\phi_{enc}, \theta, \phi, \psi, \omega\}$:
29:         $\Theta \leftarrow \Theta - \eta\nabla_\Theta\mathcal{J}$.
30: **end while**

---

**Phase 2: Coupled Continuous-Time Evolution.** The core innovation lies in the joint evolution of individual and global states (Steps 11-16). We define an augmented state vector $\mathbf{H}(t)$ that concatenates all individual latent states $\mathbf{h}_i(t)$ with the global context $\mathbf{h}_{glob}(t)$. The dynamics are governed by a **Coupled Vector Field**, where the derivative of each entity depends on both its local state and the instantaneous global state (and vice versa). Crucially, Step 16 highlights the use of the *Adjoint Sensitivity Method*. By solving the augmented ODE backward in time, we compute gradients with $\mathcal{O}(1)$ memory cost relative to the number of integration steps, making the training scalable to long sequences.

**Phase 3: Joint Optimization.** The final objective $\mathcal{J}$ (Step 20) unifies the predictive likelihood ($\mathcal{L}_{task}$) with the generative consistency terms ($\mathcal{L}_{diff}$ and $\mathcal{L}_{KL}$). This composite loss ensures that the model not only predicts the next event accurately but also learns a smooth and theoretically valid latent trajectory that respects both the local observation constraints and the global population dynamics.

## D. Evaluation Metrics

To comprehensively evaluate the performance of CoCLD and the baselines, we employ standard ranking and retrieval metrics tailored to each specific task.

### D.1. Metrics for Next-Event Prediction (Task 1)

In the Next-Event Prediction task, the goal is to predict the correct next location or event from a candidate set $\mathcal{I}$. We use **Accuracy@k** (Acc@k) and **NDCG@k**.

- **Accuracy@k (Acc@k):** Also referred to as Hit Ratio (HR@K). It measures the proportion of test cases where the ground-truth item appears in the top-$K$ predicted list.

$$\text{Accuracy@k} = \frac{1}{N} \sum_{i=1}^{N} \mathbb{I}(\text{rank}_i \leq k) \tag{10}$$

  where $N$ is the number of test sequences, $\text{rank}_i$ is the rank of the ground-truth item for the $i$-th sequence, and $\mathbb{I}(\cdot)$ is the indicator function.

- **NDCG@K (Normalized Discounted Cumulative Gain):** This metric accounts for the position of the hit. It gives higher scores if the ground-truth item is ranked higher in the list.

$$\text{NDCG@k} = \frac{1}{N} \sum_{i=1}^{N} \frac{\mathbb{I}(\text{rank}_i \leq k)}{\log_2(\text{rank}_i + 1)} \tag{11}$$

  We report results for $K \in \{5, 10\}$ to assess performance at different cutoff thresholds.

### D.2. Metrics for Trajectory Generation (Task 2)

Following prior practice (Liu et al., 2018; Zhang et al., 2023; Zhu et al., 2023), we quantify the fidelity of generated trajectories by measuring the Jensen–Shannon divergence (JSD) between distributions induced by real and generated data. Specifically, we report two complementary metrics: *Density Error* and *Trip Error*, which evaluate (i) global spatial density and (ii) user-level trip patterns via start–end correlations, respectively. Since our model can intentionally alter trajectory patterns under control signals, we avoid direct pattern-matching metrics and instead compare distributional statistics. For robustness, we repeat the generation/evaluation for 10 independent runs and report the averaged JSD scores (lower is better).

**Jensen–Shannon divergence.** Let $P$ and $Q$ denote two discrete probability distributions. We use the Jensen–Shannon divergence (JSD) to measure their discrepancy:

$$\text{JSD}(P\|Q) = \frac{1}{2}D_{\text{KL}}(P\|M) + \frac{1}{2}D_{\text{KL}}(Q\|M), \quad M = \frac{1}{2}(P + Q), \tag{12}$$

where $D_{\text{KL}}(\cdot\|\cdot)$ is the KL divergence. Lower JSD indicates closer agreement in distributional statistics.

For each city, we partition the spatial extent into a $16 \times 16$ grid. We map each GPS point to its corresponding grid cell and construct empirical distributions from either real trajectories or generated trajectories.

**Density Error (global spatial distribution).** We build a grid-frequency matrix (histogram) over all trajectory points, and normalize it into a probability distribution. Let $M^{\text{real}}$ and $M^{\text{gen}}$ be the resulting distributions for real and generated data. We define:

$$\text{DensityError} = \text{JSD}\big(M^{\text{gen}}, M^{\text{real}}\big). \tag{13}$$

This metric captures the global similarity of geographic occurrences across the entire city.

**Trip Error (start–end correlation).** To evaluate user-level trip patterns, we consider the joint distribution of start and end locations. For each trajectory, we record the grid cell of its start point and end point, and form an empirical distribution over (start-cell, end-cell) pairs for real and generated sets. Trip Error is computed as the JSD between these two joint distributions:

$$\text{TripError} = \text{JSD}\big(P_{\text{S,E}}^{\text{gen}}, P_{\text{S,E}}^{\text{real}}\big). \tag{14}$$

### D.3. Metrics for Sequential Behavior Modeling (Task 3)

For Task 3 (Amazon Review datasets), we follow the standard evaluation protocol for sequential recommendation.

- **Recall@k:** In sequential next-item prediction, Recall@k is typically defined for each user/test case $u$ as:

$$\text{Recall@k}(u) = \frac{\left|\hat{R}_{u,k} \cap G_u\right|}{|G_u|} \tag{15}$$

  where $G_u$ is the set of ground-truth relevant items (e.g., the held-out next item(s) or test interactions), and $\hat{R}_{u,k}$ is the model's Top-$k$ recommended items. The overall Recall@k is averaged over users/test cases:

$$\text{Recall@k} = \frac{1}{|U|} \sum_{u \in U} \text{Recall@k}(u) \tag{16}$$

  If there is exactly one ground-truth item per test case ($|G_u| = 1$), then Recall@k reduces to Hit Rate@k (1 if the true item appears in the Top-$k$, otherwise 0).

- **Mean Reciprocal Rank (MRR):** MRR indicates the average reciprocal rank of the ground-truth item. It focuses on how early the correct item appears in the recommendation list, which is crucial for user experience.

$$\text{MRR} = \frac{1}{N} \sum_{i=1}^{N} \frac{1}{\text{rank}_i} \tag{17}$$

  If the ground-truth item is not in the top-$K$ list, its reciprocal rank is treated as 0. MRR is generally more sensitive to the top-1 accuracy than NDCG.

## E. Additional Implementation Details

**Optimization.** Unless otherwise specified, we train for 100 epochs with batch size 128 using Adam (lr $= 10^{-3}$, weight decay $= 10^{-3}$) and early stopping (patience set to 5). We set the random seed to 2024 by default and report the mean performance over 5 runs.

**Diffusion-based Latent Interpolator.** We use a $K$-step diffusion process with $K = 5$ denoising steps. We adopt a linear noise schedule with noise scale 0.001, and clamp the noise range by `noise_min=5e−4` and `noise_max=5e−3`. We set the diffusion prediction target to `mean_type=x0` and enable timestep reweighting. The denoising network uses a DNN with hidden dimensions 300, timestep embedding size 10, and `tanh` activation. At inference time, we set the number of forward diffusion steps to 0 and disable sampling noise unless stated otherwise.

**Coupled Neural ODE Solver.** We represent individual and global states as continuous-time embeddings and integrate the coupled vector fields with a Runge–Kutta (RK4) solver. The ODE step size is set to $\Delta t{=}0.2$. The number of ODE function evaluations ($N_{ODE}$) is treated as a tunable budget parameter and is fixed across methods in each task for fair comparison. When supported, we employ the adjoint sensitivity method to reduce memory usage. We optionally enable joint optimization for ODE backward (the `--joint` flag) and control graph regularization by $\alpha$. We can enable node-level dropout with keep probability $p = 0.8$.

**Task-specific Configurations.** For next-event prediction, we report Acc@$\{5, 10\}$ and NDCG@$\{5, 10\}$. For trajectory generation, we follow prior work and report the Jensen–Shannon divergence on density and trip distributions. For sequential behavior modeling, we use a negative sampling size of 100 for evaluation (or full negatives when specified) and report Recall/NDCG at standard cutoffs. The backbone decoder uses a lightweight Transformer-style model with 2 layers and 2 attention heads, hidden size 64, FFN inner size 256, GELU activation, and dropout 0.5 on both hidden states and attention weights. We set layer normalization $\epsilon = 10^{-12}$ with initializer range 0.02. We use 2 temporal slices for constructing time-dependent graphs/states, and evaluate with 100 negative samples per user unless using full negatives.

# F. Proof of Proposition, Theorem, and Lemma

## F.1. Proof of Proposition 3.1

Proposition 3.1 (Consistency of Diffusion Interpolation): Minimizing the ELBO ensures $q_\theta$ converges to the true conditional latent posterior.

*Proof.* Let $p(\mathbf{z}_0^\tau|\mathcal{S}_i)$ be the true conditional posterior distribution of the latent state at query time $\tau$, and $q_\theta(\hat{\mathbf{z}}_0^\tau|\mathcal{S}_i)$ be the variational distribution parameterized by our time-guided diffusion model. We aim to show that minimizing the diffusion loss $\mathcal{L}_{diff}$ is equivalent to minimizing the Kullback-Leibler (KL) divergence $D_{KL}(p\|q_\theta)$.

**Step 1: Variational Lower Bound Decomposition.** The log-evidence of the latent state can be decomposed using the standard variational principle:

$$\log p(\mathbf{z}_0^\tau|\mathcal{S}_i) = \text{ELBO}(\theta; \mathcal{S}_i, \tau) + D_{KL}(q_\theta(\hat{\mathbf{z}}_0^\tau|\mathcal{S}_i)\|p(\mathbf{z}_0^\tau|\mathcal{S}_i)). \tag{18}$$

In the context of diffusion probabilistic models, the ELBO for the latent trajectory $\mathbf{z}_{0:K}^\tau$ is expressed as:

$$\mathcal{L}_{vlb} = \mathbb{E}_q\left[\log p(\mathbf{z}_K^\tau) + \sum_{k=1}^{K} \log \frac{p_\theta(\mathbf{z}_{k-1}^\tau|\mathbf{z}_k^\tau)}{q(\mathbf{z}_k^\tau|\mathbf{z}_{k-1}^\tau)}\right], \tag{19}$$

where $q(\mathbf{z}_k^\tau|\mathbf{z}_{k-1}^\tau)$ is the fixed forward diffusion kernel.

**Step 2: Equivalence to Denoising Score Matching.** As established by (Ho et al., 2020), the functional form of the variational bound can be simplified. Specifically, the KL divergence between the Gaussian transitions in the reverse and forward chains is proportional to the mean-squared error of the noise prediction:

$$\mathcal{L}_{vlb} \propto \sum_{k=1}^{K} \mathbb{E}_{q(\mathbf{z}_k^\tau|\mathbf{z}_0^\tau)}\left[D_{KL}(q(\mathbf{z}_{k-1}^\tau|\mathbf{z}_k^\tau,\mathbf{z}_0^\tau)\|p_\theta(\mathbf{z}_{k-1}^\tau|\mathbf{z}_k^\tau))\right] \tag{20}$$

$$\cong \sum_{k=1}^{K} \gamma_k \mathbb{E}_{\mathbf{z}_0^\tau,\epsilon}\left[\|\epsilon - \epsilon_\theta(\sqrt{\bar{\alpha}_k}\mathbf{z}_0^\tau + \sqrt{1-\bar{\alpha}_k}\epsilon, k, \tau, \mathcal{S}_i)\|^2\right], \tag{21}$$

where $\gamma_k$ is a weight constant and $\epsilon \sim \mathcal{N}(\mathbf{0}, \mathbf{I})$. This identifies our diffusion training objective $\mathcal{L}_{diff}$ as a reweighted version of the variational lower bound.

**Step 3: Consistency in the Continuous Limit.** Under the assumption of universal function approximation for $\epsilon_\theta$, the global minimum of $\mathcal{L}_{diff}$ is achieved if and only if:

$$\epsilon_\theta(\mathbf{z}_k, k, \tau, \mathcal{S}_i) = \mathbb{E}_q[\epsilon|\mathbf{z}_k, \tau, \mathcal{S}_i] = -\sqrt{1-\bar{\alpha}_k}\nabla_{\mathbf{z}_k} \log p(\mathbf{z}_k^\tau|\mathcal{S}_i). \tag{22}$$

This indicates that the learned score function matches the score of the true conditional density. As the number of diffusion steps $K \to \infty$ and the step size $\beta_k \to 0$, the reverse SDE (Stochastic Differential Equation) converges to the reverse temporal dynamics of the true posterior.

**Conclusion.** Since $\log p(\mathbf{z}_0^\tau|\mathcal{S}_i)$ is independent of $\theta$, maximizing the ELBO (or equivalently minimizing $\mathcal{L}_{diff}$) directly minimizes $D_{KL}(q_\theta\|p)$. Thus, the reconstructed latent state $\hat{\mathbf{z}}_0^\tau$ is a statistically consistent estimator of the true underlying latent state conditioned on the sparse event sequence $\mathcal{S}_i$. $\square$

### F.2. Proof of Theorem 3.2

Theorem 3.2 (Existence and Uniqueness of Coupled Dynamics): Given Lipschitz continuous vector fields $f_\phi$ and $g_\psi$, there exists a unique solution $(\mathbf{h}ind(t), \mathbf{h}glob(t))$ over $[0, T]$.

*Proof.* We define the augmented state vector $\mathbf{H}(t) = [\mathbf{h}_{ind}(t), \mathbf{h}_{glob}(t)]^\top \in \mathbb{R}^{d_{ind}+d_{glob}}$. The coupled system in Equation (3) and Equation (4) can be rewritten as a single non-autonomous ODE:

$$\frac{d\mathbf{H}(t)}{dt} = \mathbf{F}(\mathbf{H}(t), t), \quad \text{where}, \quad \mathbf{F} = [f_\phi, g_\psi]^\top. \tag{23}$$

i) According to the Lipschitz Continuity (Hager, 1979; Gouk et al., 2021): Since $f_\phi$ and $g_\psi$ are parameterized by multi-layer perceptrons (MLPs) with smooth activation functions (e.g., Tanh or Softplus) and bounded weights $W$, the Jacobian $\nabla_\mathbf{H}\mathbf{F}$ is bounded. Thus, there exists a constant $L > 0$ such that $\|\mathbf{F}(\mathbf{H}_1, t) - \mathbf{F}(\mathbf{H}_2, t)\| \leq L\|\mathbf{H}_1 - \mathbf{H}_2\|$ for all $\mathbf{H}_1, \mathbf{H}_2 \in \mathbb{R}^d$.

ii) Picard-Lindelöf Theorem: According to the Picard-Lindelöf theorem (Nevanlinna, 1989; Siegmund et al., 2016), if $\mathbf{F}$ is Lipschitz continuous in $\mathbf{H}$ and continuous in $t$, then for any initial condition $\mathbf{H}(t_0) = \mathbf{z}_0^\tau$, there exists a unique solution $\mathbf{H}(t)$ on a closed interval containing $t_0$.

iii) Global Existence: Since the vector fields are defined by neural networks with sub-linear growth (due to bounded weights and saturating activations), the solution does not blow up in finite time, ensuring uniqueness over the entire interval $[0, T]$.

$\square$

### F.3. Proof of Lemma 3.3

Lemma 3.3 (Stability of Latent Trajectories): The discrepancy at $t_{target}$ is bounded by $\|\mathbf{h}_{ind} - \hat{\mathbf{h}}_{ind}\| \leq \epsilon \exp(L_f \Delta t)$.

*Proof.* Let $\mathbf{h}(t)$ be the trajectory starting from the true state $\mathbf{z}_0$ and $\hat{\mathbf{h}}(t)$ be the trajectory starting from the reconstructed state $\hat{\mathbf{z}}_0$. Let $e(t) = \|\mathbf{h}(t) - \hat{\mathbf{h}}(t)\|$. The integral forms of the dynamics are:

$$\mathbf{h}(t) = \mathbf{z}_0 + \int_\tau^t f_\phi(\mathbf{h}(s), s)ds, \quad \hat{\mathbf{h}}(t) = \hat{\mathbf{z}}_0 + \int_\tau^t f_\phi(\hat{\mathbf{h}}(s), s)ds, \tag{24}$$

Subtracting the two and applying the triangle inequality:

$$e(t) \leq \|\mathbf{z}_0 - \hat{\mathbf{z}}_0\| + \int_\tau^t \|f_\phi(\mathbf{h}(s), s) - f_\phi(\hat{\mathbf{h}}(s), s)\|ds, \tag{25}$$

Using the Lipschitz property (Balan et al., 2017; Cobzaş et al., 2019), having $\|f(\mathbf{h}) - f(\hat{\mathbf{h}})\| \leq L_f\|\mathbf{h} - \hat{\mathbf{h}}\|$:

$$e(t) \leq \epsilon + \int_\tau^t L_f e(s)ds, \tag{26}$$

By Grönwall's Inequality (Dannan, 1985), if $e(t) \leq \epsilon + \int L_f e(s)ds$, then:

$$e(t) \leq \epsilon \exp\left(\int_\tau^t L_f ds\right) = \epsilon \exp(L_f(t - \tau)), \tag{27}$$

This proves that the interpolation error $\epsilon$ propagates at most exponentially with the Lipschitz constant of the dynamics. $\square$

## F.4. Proof of Theorem 3.4

Theorem 3.4 (Universal Approximation of Coupled Continuous Dynamics): CoCLD can approximate any continuous joint trajectory $(\mathcal{H}_{ind}, \mathcal{H}_{glob})$ on $[0, T]$.

*Proof.* Let $\mathcal{H}^*(t) = [\mathcal{H}_{ind}^*(t), \mathcal{H}_{glob}^*(t)]^\top \in \mathbb{R}^{d+D}$ be a target joint continuous trajectory defined on $t \in [0, T]$. Our goal is to prove that for any $\delta > 0$, there exists a CoCLD configuration $\mathbf{F}_\Theta$ such that $\sup_{t \in [0,T]} \|\mathbf{H}(t) - \mathcal{H}^*(t)\| < \delta$.

**Step 1: Representation via Target Vector Field.** According to the embedding theorem for dynamical systems (Cresson & Darses, 2007), any continuous trajectory $\mathcal{H}^*(t)$ in a compact domain can be viewed as the solution to a non-autonomous ODE:

$$\frac{d\mathcal{H}^*(t)}{dt} = \mathbf{V}^*(\mathcal{H}^*(t), t), \tag{28}$$

where $\mathbf{V}^* : \mathbb{R}^{d+D} \times [0, T] \to \mathbb{R}^{d+D}$ is a continuous vector field. Due to the coupling nature of CoCLD, we decompose $\mathbf{V}^*$ into two components: $\mathbf{V}^* = [\mathbf{v}_{ind}^*, \mathbf{v}_{glob}^*]^\top$.

**Step 2: Approximation of the Vector Field.** The CoCLD framework defines the joint vector field $\mathbf{F}_\Theta(\mathbf{H}, t)$ as:

$$\mathbf{F}_\Theta(\mathbf{H}, t) = \begin{bmatrix} f_\phi(\mathbf{h}_{ind}(t), \psi(\mathbf{h}_{glob}(t))) \\ g_\psi(\mathbf{h}_{glob}(t), \rho(\{\mathbf{h}_{ind}(t)\})) \end{bmatrix}. \tag{29}$$

By the Universal Approximation Theorem for MLPs (Hornik et al., 1989; Augustine, 2024), for any continuous functions $\mathbf{v}_{ind}^*$ and $\mathbf{v}_{glob}^*$, and any $\eta > 0$, there exist neural networks $f_\phi, g_\psi$ and coupling functions $\psi, \rho$ such that:

$$\sup_{\mathbf{H} \in \mathcal{K}, t \in [0,T]} \|f_\phi(\mathbf{h}_{ind}, \psi(\mathbf{h}_{glob})) - \mathbf{v}_{ind}^*(\mathbf{h}_{ind}, \mathbf{h}_{glob}, t)\| < \eta, \tag{30}$$

$$\sup_{\mathbf{H} \in \mathcal{K}, t \in [0,T]} \|g_\psi(\mathbf{h}_{glob}, \rho(\{\mathbf{h}_{ind}\})) - \mathbf{v}_{glob}^*(\mathbf{h}_{ind}, \mathbf{h}_{glob}, t)\| < \eta, \tag{31}$$

where $\mathcal{K}$ is a compact set containing the trajectories. Consequently, $\|\mathbf{F}_\Theta(\mathbf{H}, t) - \mathbf{V}^*(\mathbf{H}, t)\| < \sqrt{2}\eta$.

**Step 3: Trajectory Convergence via Grönwall's Inequality.** Let $\mathbf{H}(t)$ be the trajectory generated by CoCLD starting from $\mathbf{H}(0) = \mathcal{H}^*(0)$. The discrepancy $e(t) = \|\mathbf{H}(t) - \mathcal{H}^*(t)\|$ satisfies the integral inequality:

$$e(t) = \left\| \int_0^t (\mathbf{F}_\Theta(\mathbf{H}(s), s) - \mathbf{V}^*(\mathcal{H}^*(s), s)) \, ds \right\| \tag{32}$$

$$\leq \int_0^t \|\mathbf{F}_\Theta(\mathbf{H}(s), s) - \mathbf{F}_\Theta(\mathcal{H}^*(s), s)\| ds + \int_0^t \|\mathbf{F}_\Theta(\mathcal{H}^*(s), s) - \mathbf{V}^*(\mathcal{H}^*(s), s)\| ds. \tag{33}$$

Applying the Lipschitz continuity (Shang et al., 2021) of the CoCLD vector field (with constant $L$) and the approximation bound from Step 2:

$$e(t) \leq \int_0^t Le(s)ds + \sqrt{2}\eta t \leq \int_0^t Le(s)ds + \sqrt{2}\eta T. \tag{34}$$

By Grönwall's Inequality (Dannan, 1985; Feng et al., 2024), we obtain:

$$e(t) \leq (\sqrt{2}\eta T) \exp(Lt). \tag{35}$$

To satisfy the target precision $\delta$, we choose $\eta$ such that $\eta < \frac{\delta}{\sqrt{2}T \exp(LT)}$. Thus, $\sup_{t \in [0,T]} e(t) < \delta$, completing the proof. $\square$

# G. Additional Experimental Result

## G.1. Ablation Result on the Amazon Dataset

We examine the impact of key components on the Amazon dataset, as shown in Figure 11. The **w/o Coupling** variant, which removes the global ODE $g_\psi$, suffers a significant performance drop (Recall@10 decreases by approx. $25\%$). This confirms that individual behaviors are not isolated but are intrinsically modulated by population-level trends. Similarly, the **w/o Diffusion** variant, using linear interpolation instead, degrades performance by over $40\%$. This validates that simple heuristic interpolation is insufficient for irregular sequences, whereas our time-guided diffusion interpolator provides high-quality latent initialization.

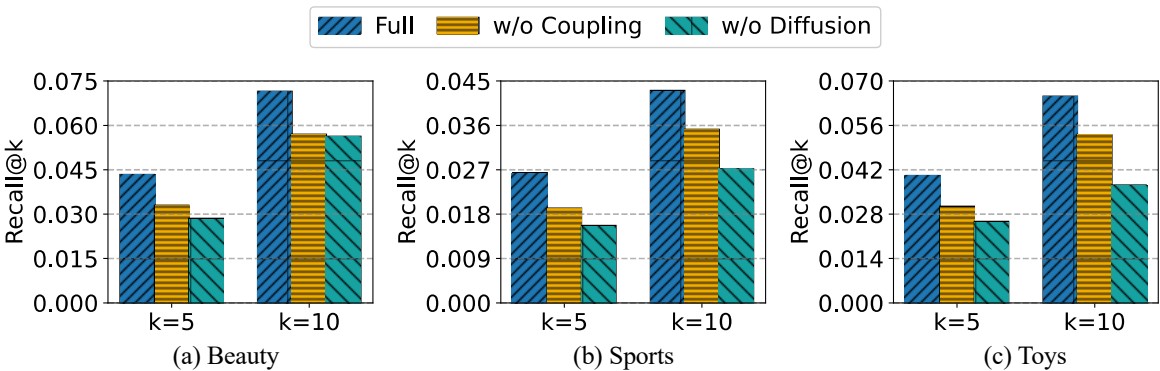

*Figure 11.* Ablation study on three sub-categories of the Amazon Review dataset.

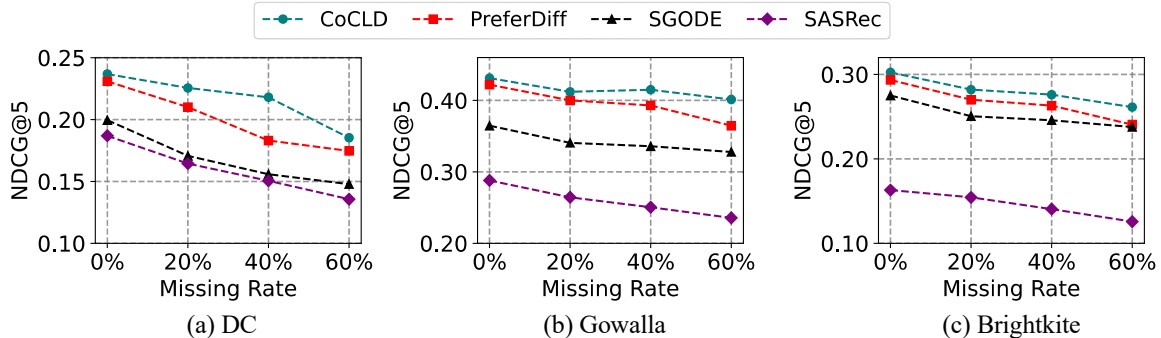

*Figure 12.* Performance degradation under increasing data sparsity ratios on the DC, Gowalla, and Brightkite datasets.

## G.2. Robustness to Sparsity on the Remaining Three Datasets

Figure 12 shows the performance degradation under increasing data sparsity ratios on the DC, Gowalla, and Brightkite datasets. Although PreferDiff (the most competitive model) also employs a diffusion mechanism, but there are three fundamental differences:

i) **Continuous vs. Static Time Modeling:** PreferDiff incorporates temporal information primarily as static condition embeddings. It lacks an explicit mechanism to model the continuous state evolution over irregular intervals. In contrast, CoCLD leverages Neural ODEs to perform continuous integration, ensuring that the latent state trajectory is mathematically consistent with the actual time lapse, rather than just position-aware.

ii) **Synergy of Interpolation and Dynamics:** PreferDiff treats sequence modeling as a direct conditional generation task, which can suffer from mode collapse when supervision signals are sparse. CoCLD adopts a "Interpolate-then-Evolve" paradigm. We use diffusion solely as a robust latent interpolator to reconstruct the initial state, while delegating the temporal evolution to the deterministic ODE solver. This separation of concerns improves stability (as shown in Lemma 3.3).

iii) **Global-Individual Coupling:** Standard diffusion baselines typically model user sequences in isolation or via implicit attention. They fail to explicitly capture how population-level distributional shifts (e.g., aggregate mobility trends) modulate individual trajectories. CoCLD's coupled mechanism ($g_\psi$) allows global dynamics to rectify individual predictions, effectively reducing variance when individual data is scarce.

## G.3. Visual Analysis of Trajectory Generation

To further evaluate the generative quality of CoCLD, we provide high-resolution visualizations of the generated trajectories on the road networks of Chengdu (Figure 13) and Xi'an (Figure 14).

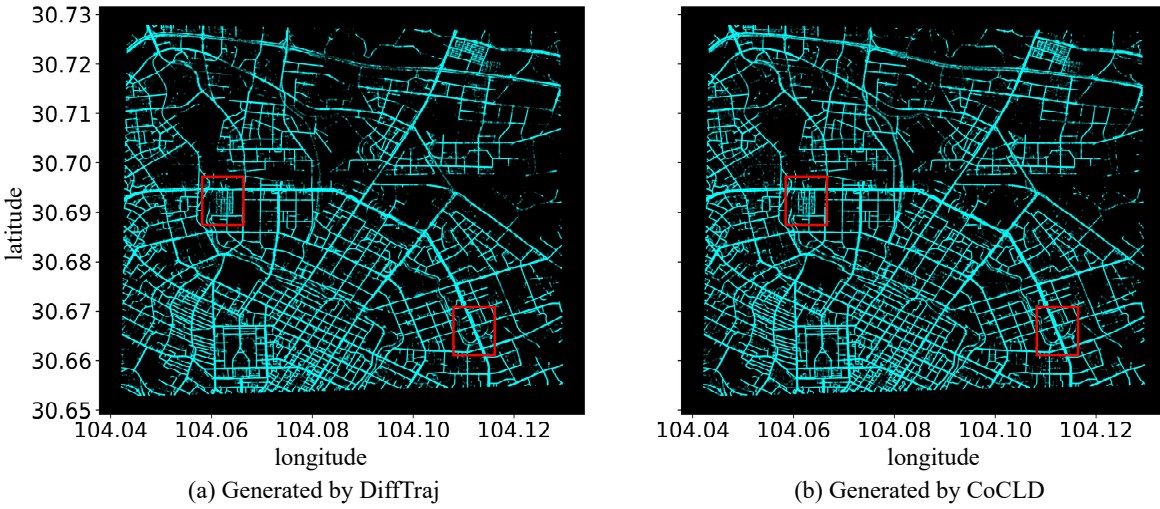

(a) Generated by DiffTraj             (b) Generated by CoCLD

*Figure 13.* Visualization of generated mobility trajectories on the Chengdu road network. Our CoCLD model generates trajectories with significantly higher adherence to the actual road geometry and less spatial noise.

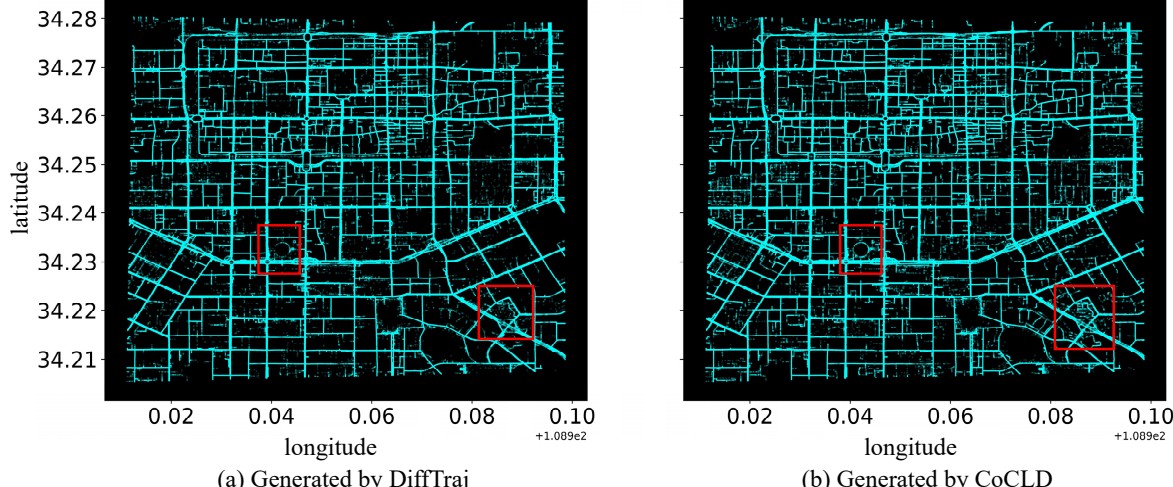

(a) Generated by DiffTraj             (b) Generated by CoCLD

*Figure 14.* Visualization of generated mobility trajectories on the Xi'an road network. CoCLD accurately reconstructs the intensity of urban hotspots and the connectivity of the road network, whereas the DiffTraj exhibits some over-smoothing or missing key clusters.

**Spatial Fidelity and Road Constraints.** As illustrated in Figure 13, CoCLD demonstrates superior spatial fidelity compared to baseline generative models. While DiffTraj produces some "off-road" points or trajectories that unrealistically cross buildings (due to the lack of physical constraints), CoCLD generates paths that strictly adhere to the underlying road network. This is achieved by the continuous-time ODE evolution, which treats the latent space as a smooth manifold reflecting the city's topological structure.

**Global Distribution and Hotspot Consistency.** Figure 14 presents a trajectory comparison on the Xi'an dataset. The trajectories generated by CoCLD preserve the semantic hotspots (e.g., commercial centers and transportation hubs) with remarkable consistency relative to the ground truth. This superiority stems from our Individual-Global Coupling mechanism: by aligning individual latent dynamics with the population-level distributional shifts ($h_{glob}$), CoCLD ensures that generated agents naturally converge toward high-density areas during peak hours, replicating the collective intelligence observed in real-world urban flows.

## G.4. Interpretability Analysis of the Proposed CoCLD Model

To improve the transparency of CoCLD, we analyze the learned latent representations by visualizing the distribution of several interpretable, behavior-related factors. Specifically, for each user we extract: (i) the *individual–group coupling score* (quantifying how strongly an individual's latent dynamics are aligned with the population-level process), (ii) the *temporal asynchrony index* (capturing the degree of irregularity/misalignment in interaction times), and (iii) a *responsiveness* measure (reflecting how sensitively the user state reacts to preference-evolution signals). We then plot the empirical distributions and perform clustering in this feature space. As shown in Figure 15, taking NYC as an example, the three factors exhibit multi-modal structures, and the resulting clusters correspond to distinct behavioral regimes, indicating that CoCLD does not collapse users into a single homogeneous representation but instead organizes them into meaningful groups.

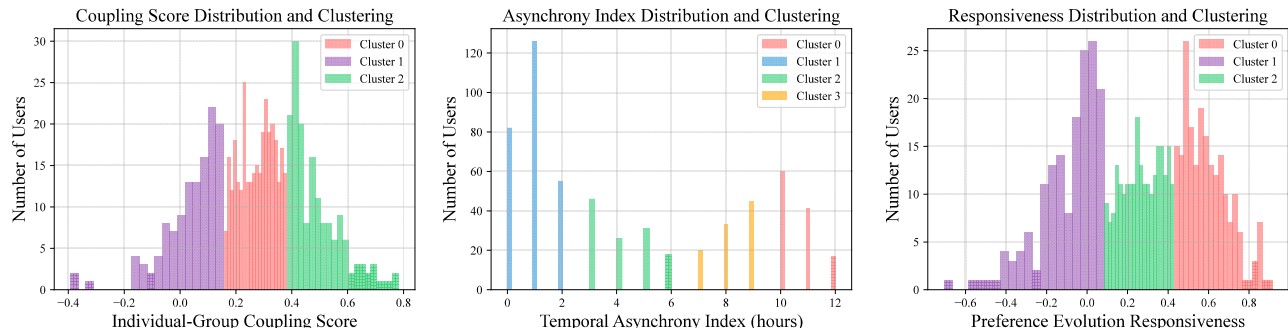

*Figure 15.* Interpretability of feature distribution learned by the CoCLD model (**NYC dataset**) from three perspectives: (a) coupling score distribution; (b) asynchrony index distribution; (c) responsiveness distribution.

We further project the learned features into low-dimensional spaces for inspection. Figure 16 reports both a PCA-based 2D visualization and a 3D scatter plot in the original factor space. The clusters form compact regions with clear boundaries, suggesting that the learned representation is highly separable and that different interaction patterns can be well distinguished by the coupling/asynchrony/responsiveness factors. This empirical separability provides qualitative evidence that CoCLD captures diverse individual–global mixed interaction patterns in a structured and interpretable manner.

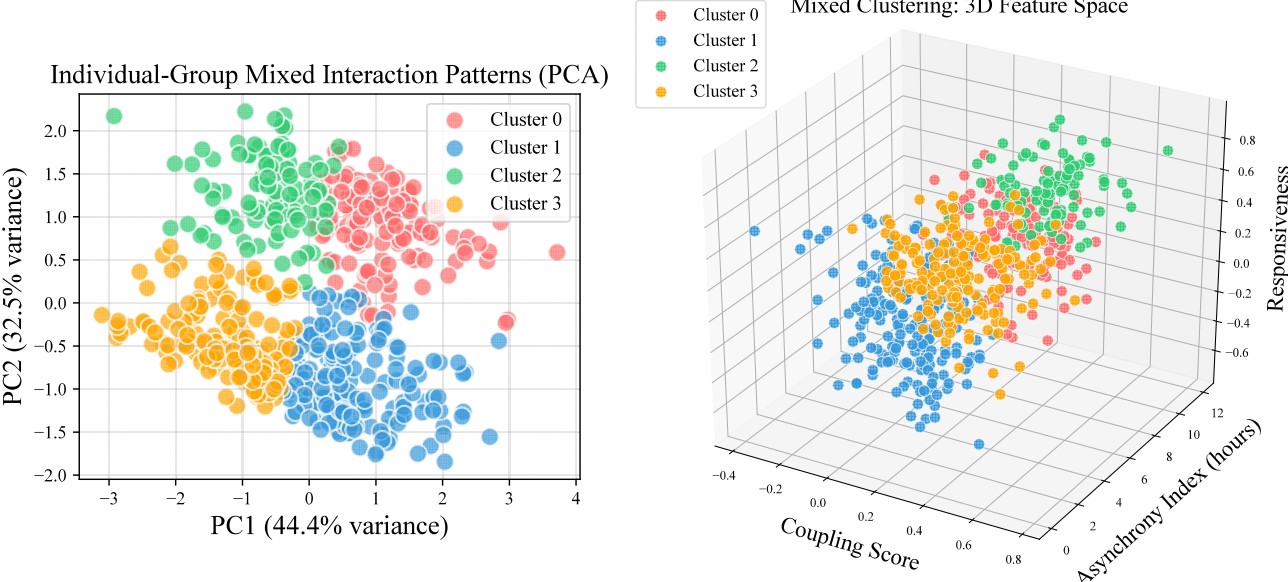

*Figure 16.* Interpretability of feature distribution learned by the CoCLD model. PCA 2D scatter and 3D scatter plots (**NYC dataset**).

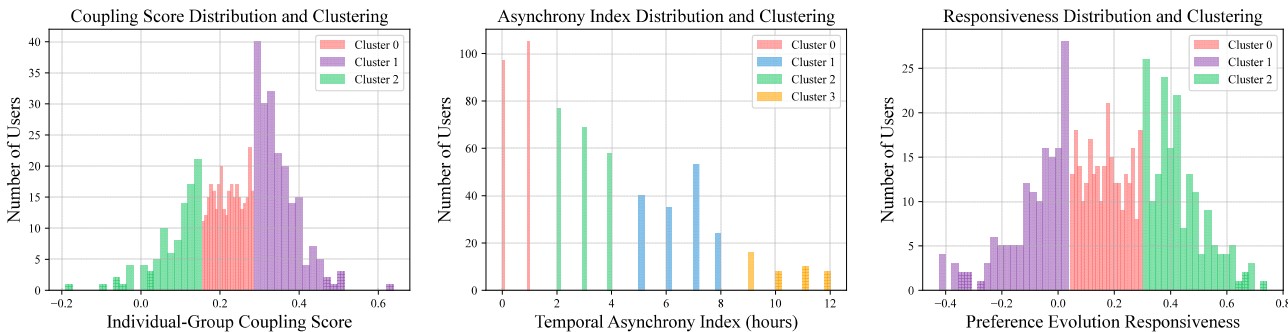

*Figure 17.* Interpretability of feature distribution learned by the CoCLD model (**IST dataset**) from three perspectives: (a) coupling score distribution; (b) asynchrony index distribution; (c) responsiveness distribution.

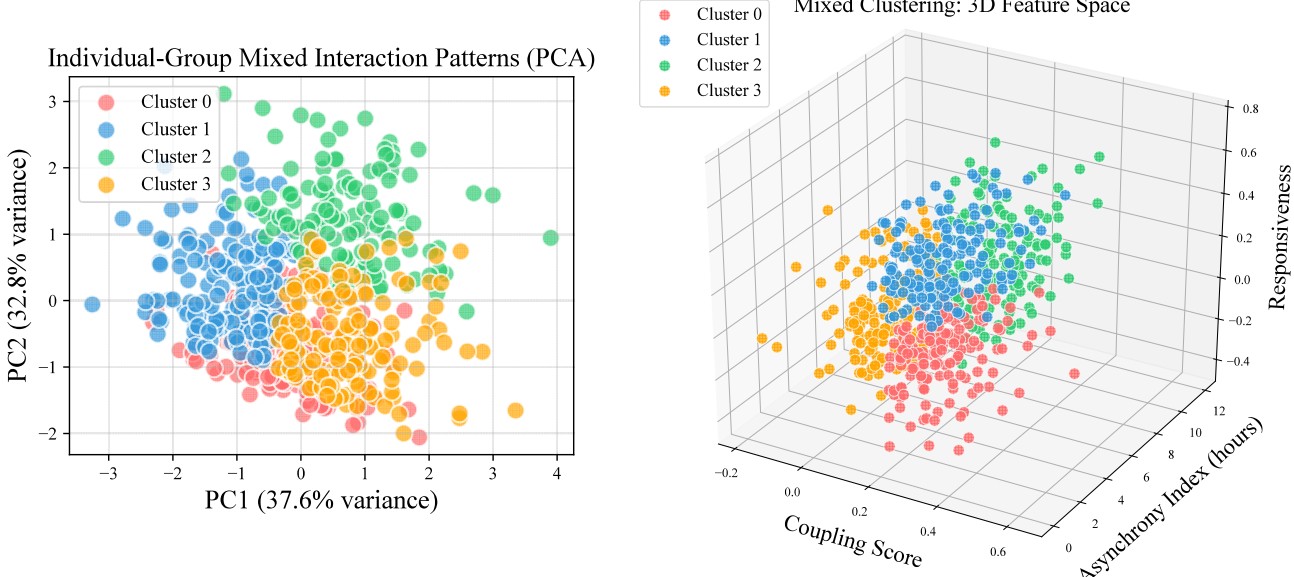

*Figure 18.* Interpretability of feature distribution learned by the CoCLD model. PCA 2D scatter and 3D scatter plots (**IST dataset**).

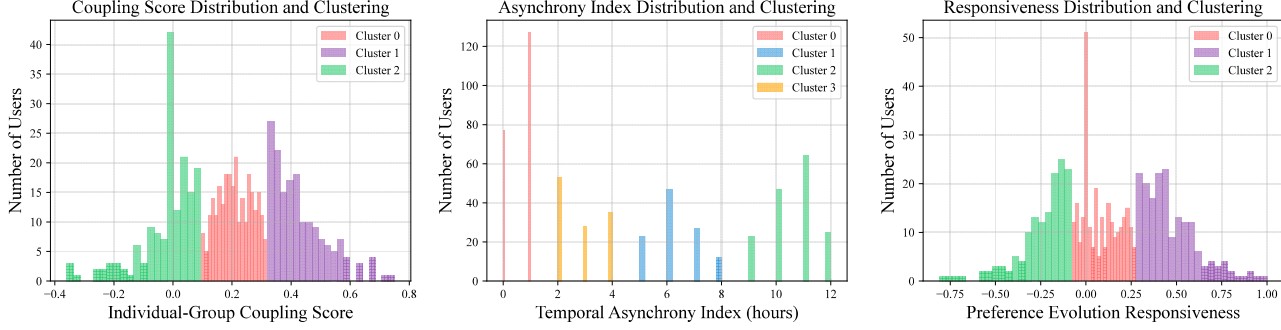

*Figure 19.* Interpretability of feature distribution learned by the CoCLD model (**DC dataset**) from three perspectives: (a) coupling score distribution; (b) asynchrony index distribution; (c) responsiveness distribution.

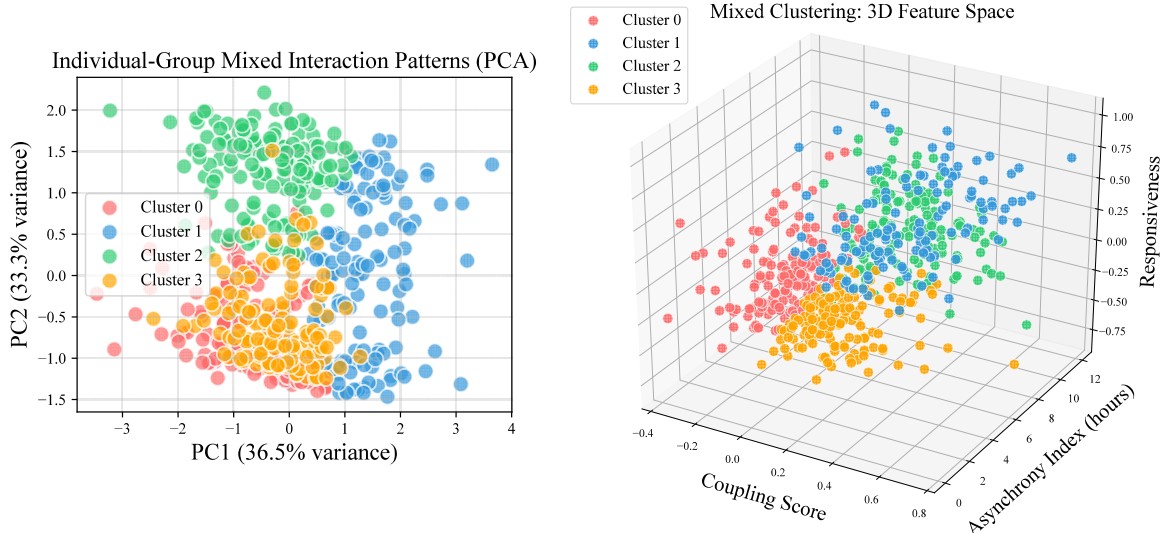

*Figure 20.* Interpretability of feature distribution learned by the CoCLD model. PCA 2D scatter and 3D scatter plots (**DC dataset**).

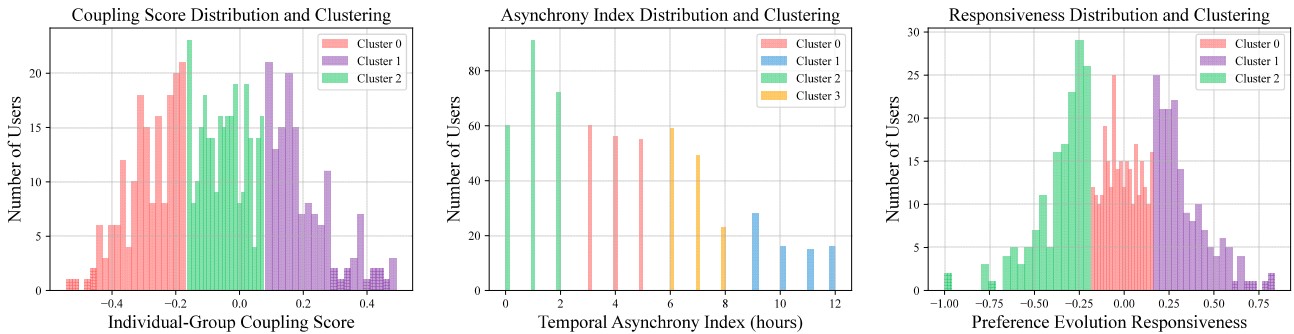

*Figure 21.* Interpretability of feature distribution learned by the CoCLD model (**Gowalla dataset**) from three perspectives: (a) coupling score distribution; (b) asynchrony index distribution; (c) responsiveness distribution.

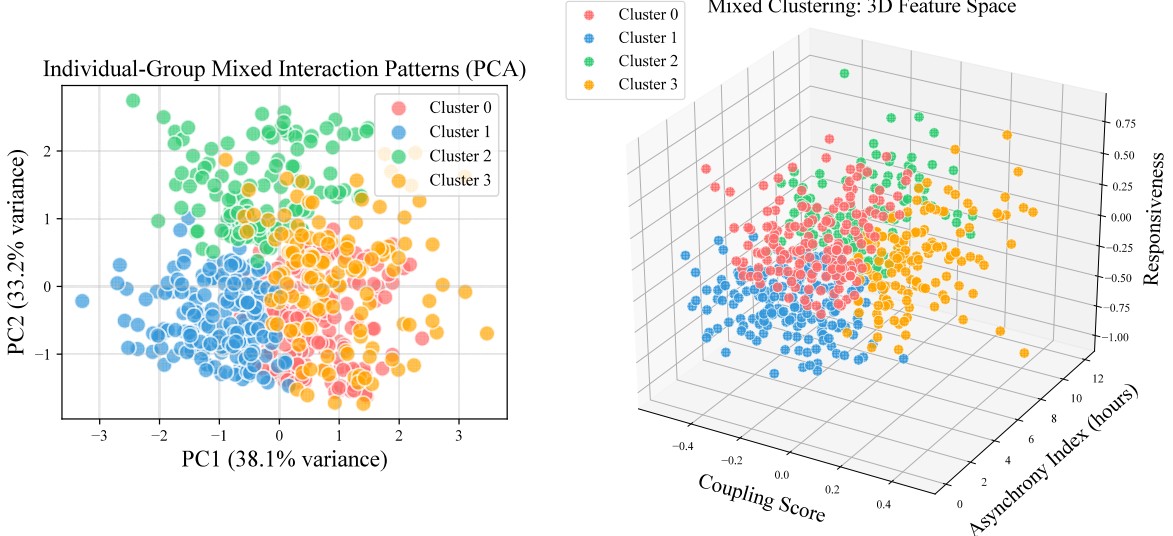

*Figure 22.* Interpretability of feature distribution learned by the CoCLD model. PCA 2D scatter and 3D scatter plots (**Gowalla dataset**).

