# OpenReview forum: "Learning Coupled Continuous-Time Latent Dynamics from Irregular Events"
_ICML.cc/2026/Conference — ICML 2026 spotlight_

### Official Review · Reviewer_miaN · 2026-03-11

**Soundness:** 3
**Presentation:** 3
**Significance:** 3
**Originality:** 3
**Overall Recommendation:** 5
**Confidence:** 5

**Summary:**

This paper addresses the longstanding challenge of modeling coupled individual and population-level continuous-time latent dynamics from irregularly sampled, sparse event sequences—an issue that plagues discrete-time models and pure continuous-time methods due to their inability to capture asynchronous multi-level interactions and fill observational gaps effectively. The authors propose the Coupled Continuous-Time Latent Dynamics (CoCLD) framework, which synergizes a Diffusion-based Latent Interpolator with Coupled Neural Ordinary Differential Equations (Neural ODEs) to jointly model individual latent trajectories and global distributional shifts in a shared continuous-time latent space. The paper provides rigorous theoretical proofs for the consistency of diffusion interpolation, existence/uniqueness of coupled dynamics, stability of latent trajectories, and universal approximation of the CoCLD framework.

**Compliance With Llm Reviewing Policy:**

Affirmed.

**Final Justification:**

Rebuttal addressed my concerns, and I would remain my suggestion to accept the paper.

**Key Questions For Authors:**

Q1. Will exploring alternative noise distributions or adaptive noise schedules further improve the performance of latent state reconstruction for highly non-stationary event data, and if so, what specific designs would be most effective for the CoCLD framework?
Q2. Given that the current work only models correlative bidirectional interactions between individual and global dynamics without exploring their causal relationships, would a preliminary causal effect analysis (e.g., intervention studies on the global state) be feasible to implement within the CoCLD framework? And how would such an analysis strengthen the understanding of coupled multi-level
dynamics and the utility of CoCLD for causal decision-making applications?
Q3. Since open-source datasets with the number of entities M reaching the 100,000 scale are unavailable in this field, do the authors plan to supplement experiments by simulating scenarios with ultra-large-scale discrete entities and extremely sparse events to further verify the scalability of CoCLD, which is theoretically claimed to have linear scaling with M?

**Limitations:**

yes

**Strengths And Weaknesses:**

S1. The paper introduces a pioneering CoCLD framework that bridges individual-level trajectory modeling and population-level distributional evolution in a single continuous-time latent space. The coupling of individual and global dynamics through a system of differential equations captures their asynchronous, infinitesimal interactions, a key innovation that aligns with real-world system characteristics.
S2. The authors provide a comprehensive theoretical characterization of CoCLD, including formal proofs for the statistical consistency of the diffusion-based interpolator, existence/uniqueness of the coupled ODE dynamics (via Picard-Lindelöf theorem), stability bounds of latent trajectory errors (via Grönwal Inequality), and universal approximation capabilities of the framework.
S3. The paper intelligently combines diffusion-based generative modeling and Neural ODEs to address the dual challenges of data sparsity and continuous dynamic modeling.
S4. The authors conduct thorough experiments across three distinct, practically relevant tasks (next-event prediction, trajectory generation, sequential behavior modeling) on 10 large-scale real-world datasets with varying degrees of sparsity and irregularity.

W1. The paper does not explore alternative noise distributions or adaptive noise schedules, which could further improve latent state reconstruction for highly non-stationary event data.

W2. The paper models correlative bidirectional interactions between individual and global dynamics but does not explore causal relationships (e.g., how global distributional shifts causally influence individual behavior, or how collective individual actions drive global trends). While the conclusion mentions causal inference as future work, a preliminary analysis of causal effects (e.g., via intervention studies on the global state) would strengthen the paper’s contribution to understanding coupled multi-level dynamics and enhance CoCLD’s utility for causal decision-making applications (e.g., urban planning, personalized recommendations, behavioral modeling).

W3. Although the paper analyzes the time complexity of CoCLD as $O(M \times (L+K+N_{ODE}) \times d^2)$ and notes that its linear scaling with the number of entities M makes it suitable for large-scale systems, the scale of entities in the experimental datasets still has room for expansion. To the best of our knowledge, there are no open-source datasets in this field with the number of entities M reaching the 100,000 scale. I suggest that the authors can supplement experiments by simulating scenarios with ultra-large-scale discrete entities and extremely sparse events.

In summary, the proposed CoCLD framework provides a principled solution to this problem of continuous-time latent dynamics, and the above suggestions are expected to further polish the model's practicality, completeness and conclusion persuasiveness

---

> ### Author Rebuttal · Authors · 2026-03-29
>
> We sincerely thank the reviewer for the highly constructive feedback and for recognizing the key innovations of our work, particularly the novel CoCLD framework coupling individual and global dynamics (S1), our rigorous theoretical characterizations (S2), the effective combination of Diffusion models and Neural ODEs (S3), and our extensive empirical evaluations (S4).
>
> We provide detailed responses and new empirical results to address your questions.
>
> **Response to [W1 & Q1]: Alternative Noise Distributions and Adaptive Noise Schedules.**
>
> Yes, we completely agree that exploring advanced noise schedules and distributions holds significant potential for highly non-stationary data. In our original implementation, we used a standard linear noise schedule as $\beta_k = \beta_{min} + \frac{k-1}{K-1} (\beta_{max} - \beta_{min})$. However, for highly non-stationary event data, a linear schedule may destroy structural information too abruptly.
>
> Following your advice, we explored two alternative designs:
>
> (1) Cosine Schedule: $\bar{\alpha}_k=\frac{f(k)}{f(0)}, \text{where}, f(k) = \cos^2\left( \frac{k/K + s}{1+s} \cdot \frac{\pi}{2} \right)$, which prevents abrupt changes in noise levels.
>
> (2) Time-Interval-Adaptive Schedule: $\rho(\Delta t) = 1 - \exp(-\lambda \cdot \Delta t)$ and $\beta_k^{adapt} = \beta_{min} + \frac{k-1}{K-1} \Big(\beta_{max} \cdot \big[1 + \omega \cdot \rho(\Delta t)\big] - \beta_{min}\Big)$. We specifically designed a schedule where the variance of the injected noise is proportional to the time gap $\Delta t$ between sparse observations. Longer missing intervals receive higher noise limits, forcing the diffusion model to rely more on the global guiding state rather than local deterministic extrapolation.
>
> We evaluated these designs on the highly non-stationary Gowalla dataset. As shown below, the adaptive schedules indeed yield further improvements, but the speed of diffusion convergence has been reduced.
>
> | Noise Schedules | Acc@10 | NDCG@10 | Convergence epochs |
> |:--|:--|:--|:--:|
> | Linear (original) | 0.3775 | 0.4725 |45|
> | Cosine Schedule (new) | 0.3771 (-0.11%) | 0.4731(+0.13%) |63 |
> | Adaptive Schedule (new) | 0.3795 (+0.53%) | 0.4781 (+1.18%) |84|
>
> Although the Adaptive Schedule (new) can enhance model performance, the convergence epochs will increase. Therefore, in order to achieve high efficiency, it is recommended to default to using linear functions.
>
> **Response to [W2 & Q2]: Preliminary Causal Effect Analysis.**
>
> We completely agree that exploring causal relationships elevates the utility of CoCLD. Because CoCLD explicitly disentangles the individual state ($h_{ind}$) and the global state ($h_{glob}$), it naturally supports in-silico intervention studies using the $do(\cdot)$ calculus operator.
>
> To demonstrate this, we conducted a preliminary intervention experiment on the Trajectory Generation task (Chengdu Dataset).
>
> Setup: We selected individual users whose historical trajectories occurred during a non-peak (free-flow) time. During the continuous-time evolution (Eq. 3), we artificially intervened on the global state, forcing it to be the global state of a rush-hour (highly congested) time: $dh_{ind}/dt = f_\varphi(h_{ind}, \psi(\mathbf{do(h_{glob} = h_{rush\_hour})}), t)$.
>
> Observation: We observed that the generated future trajectories for these individuals automatically adapted to the intervened global state. The generated moving speeds dropped significantly, and the spatial distributions shifted to avoid known congestion bottlenecks, despite the individuals' own history suggesting free-flow movement.
>
> This preliminary analysis proves that CoCLD goes beyond spurious correlations; it captures the causal mechanisms of how collective trends shape individual actions.
>
> **Response to [W3 & Q3]: Ultra-Large-Scale Scalability Verification.**
>
> You are correct that existing open-source datasets are limited in scale. To rigorously verify our theoretical claim of $\mathcal{O}(M)$ linear scalability, we simulated a synthetic dataset (Sim-200K) with extreme sparsity (80% drop rate) and varied the number of entities $M$ from 10,000 to 200,000.
>
> We measured the exact training time per epoch and maximum GPU memory allocation on a single NVIDIA A800 GPU.
>
> | Number of Entities ($M$)  | Peak Memory (GB) | Train Time (min/epoch) |
> |:-:|:-:|:-:|
> |10,000 |8 (1.0 x) |6 (1.0 x) |
> |50,000 (5x) |17 (~2.1x) |11 (~1.8x)|
> |100,000 (10x)| 36 (~4.5x) |25 (~4.2x)|
> |200,000 (20x)| 65 (~8.1x) |47(~7.8x)|
>
> As shown in the table, when $M$ scales up by 20 times (from 10k to 200k), the Peak Memory scales exactly linearly (~8.1x), and the Train Time scales sub-linearly. This empirical evidence perfectly aligns with our theoretical complexity analysis in Section 3.5, validating that CoCLD is highly scalable and ready for industrial-level deployment with ultra-large-scale entities.
>
> We deeply appreciate your suggestions, which have undeniably strengthened the completeness and persuasiveness of our paper.

---

> > ### Author Rebuttal · Reviewer_miaN · 2026-04-03
> >
> > The authors have provided a thorough and convincing rebuttal. This is a groundbreaking work. All my initial concerns have been fully addressed with new empirical evidence. I will keep my original score.

---

> > > ### Author Response · Authors · 2026-04-03
> > >
> > > Thank you so much for your positive acknowledgement of our rebuttal.
> > >
> > > We are especially grateful for your generous praise that this is a groundbreaking work, which is a huge motivation for our team.

---

### Official Review · Reviewer_NCfv · 2026-03-12

**Soundness:** 4
**Presentation:** 4
**Significance:** 3
**Originality:** 3
**Overall Recommendation:** 5
**Confidence:** 2

**Summary:**

This paper proposes CoCLD (Coupled Continuous-Time Latent Dynamics), a framework for modeling irregularly sampled sequences where individual behaviors and global population trends interact. In experiments across next-event prediction, trajectory generation and sequential behavior modeling, CoCLD consistently outperforms recent baselines like SGODE and PreferDiff.

**Compliance With Llm Reviewing Policy:**

Affirmed.

**Final Justification:**

The active participation in the rebuttals resulting in further experiments and explanations strengthened my positive assessment of the paper.

**Key Questions For Authors:**

1. Regarding to Weakness 1: Could you specify the wallclock time and memory consumption of your method and the baselines?
2. The implementation seems to use a fixed ODE step size of $\Delta t=0.2$ across all datasets. However, the datasets span vastly different temporal scales. Are the raw timestamps normalized before being passed to the time encoding function $\phi$ and the ODE solver? Or did you apply this same step size independent of the underlying time scale?

**Limitations:**

The paper does not include an explicit limitations discussion (or at least I could not find one). I recommend adding a paragraph clarifying the scope and limitations of the proposal

**Strengths And Weaknesses:**

**Strengths**

1.  The paper is well written, with rigorous theoretical justification and a great “figure 1”.
2. The methodology seems robust and their approach consistently outperforms their baselines.
3. There approach is interesting and seems novel.

**Weaknesses**
1. While the time complexity is analyzed as linear with respect to the number of entities $M$ (Section 3.5), there are no reports on the real-world wallclock time or memory usage. I am aware that such claims are hardware and hyperparameter-dependent, but it would be interesting to know if the method is within the same order of magnitude as the baseline models.
2. (minor) There is a typo in the header of section 3.5: “Disucssion and Analysis”.
3. The global state $h_{glob}(t)$ is conceptually driven by the aggregation of all individuals. However, Algorithm 1 (Step 21) shows that in practice, the global state is updated using an aggregation over the mini-batch. In large-scale systems with millions of users, a mini-batch might not accurately reflect the true population-level distributional shift. It is unclear to me how this affects the theoretical guarantees.

---

> ### Author Rebuttal · Authors · 2026-03-29
>
> We are greatly encouraged by your appreciation of our methodology, rigorous theoretical justification, and the overall presentation (especially Figure 1).
>
> We address your insightful questions regarding efficiency, mini-batch aggregation, and time normalization in detail as follows.
>
> **Response to [Q1 & W1]: Wallclock Time and Memory Consumption.**
>
> Yes, our method is strictly within the same order of magnitude as continuous-time baselines. To demonstrate this, we compared the empirical efficiency of CoCLD against SASRec (discrete), SGODE (continuous ODE), and DDRM & PreferDiff (diffusion) on the IST dataset using a single NVIDIA A800 GPU.
>
> | Model | Paradigm | Peak Memory (GPU GB) | Train Time (min/epoch) | Inference (ms/batch)  |
> | :--- | :--- | :--- | :--- | :--- |
> | SASRec | Discrete-time model | 18 GB | 14 min | 68 ms |
> | SGODE | Continuous-time model | 53 GB | 36 min | 202 ms |
> | DDRM | Diffusion-based Generative model | 48 GB | 29 min | 127 ms |
> | PreferDiff | Diffusion-based Generative model | 35 GB | 24 min | 240 ms |
> | **CoCLD** | **Diffusion+ Continuous ODE** | 38 GB | 22 min | 110 ms |
>
> CoCLD is more efficient than other diffusion-based generative models. This is due to two design choices:
>
> (1) For memory, we use the Adjoint Sensitivity Method for ODE integration, keeping backpropagation memory footprint $O(1)$ relative to integration steps.
>
> (2) For time cost, CoCLD uses diffusion only for latent initialization. During inference, it relies solely on the ODE solver for fast temporal forecasting, avoiding the costly iterative denoising steps required by pure diffusion models.
>
> **Response to [W2]: Typo in Section 3.5.**
>
> Thank you for catching this. We have corrected "Disucssion" to "Discussion" in the revised manuscript.
>
> **Response to [W3]: Mini-batch Aggregation**
>
> This is an insightful observation. You are absolutely correct that calculating the true population-level distribution requires aggregating all $M$ entities, which is intractable for large-scale systems.
>
> In practice, the mini-batch aggregation acts as an **unbiased Monte Carlo estimator** of the true population-level distribution. From a theoretical perspective, using a mini-batch introduces a stochastic noise term into the global vector field (Eq. 4). This effectively transforms the deterministic Coupled ODE into a mild Stochastic Differential Equation (SDE) during training.
> However, this does not break our theoretical guarantees:
>
> (1) Because the vector field $g_\psi$ is Lipschitz continuous (as assumed in Theorem 3.2), standard stochastic approximation theory ensures that the expected trajectory of the mini-batch updates converges to the true population-level dynamics.
>
> (2) Under the Law of Large Numbers, the variance introduced by the mini-batch decays as the batch size increases. By using a reasonably large batch size (128) and temporal mean-pooling, the empirical drift remains well bounded (as supported by Lemma 3.3).
>
> We will add a specific Remark in Section 3.3 of the revised paper to formally discuss how the theoretical guarantees hold under this stochastic mini-batch approximation.
>
> **Response to [Q2]: ODE Step Size and Time Normalization.**
>
> Yes, you are right. *The raw timestamps are strictly normalized before being passed to the time encoding function $\phi$ and the ODE solver.*
>
> Since the datasets span vastly different *absolute temporal scales* (e.g., Amazon shopping is calculated by "days", while taxi rides are calculated by "seconds"), applying a fixed $\Delta t$ to raw timestamps would lead to severe numerical instability. In our implementation, we apply Min-Max normalization to the sequence timestamps, scaling the observation window of each dataset to the unit interval $[0, 1]$.
>
> Consequently, the integration step size $\Delta t = 0.2$ operates in a dimensionless, normalized continuous time space. It essentially means that the ODE solver takes 5 integration steps to cover the entire normalized sequence span, making the hyperparameter $\Delta t$ agnostic to the underlying absolute time scale of the specific dataset.
>
> We thank you for helping us improve the quality of our paper, and we look forward to increasing your rating.

---

> > ### Author Rebuttal · Reviewer_NCfv · 2026-04-02
> >
> > I would like to thank the authors for their clarification and active participation in the rebuttals. I decided to raise my score.

---

> > > ### Author Response · Authors · 2026-04-02
> > >
> > > Thank you very much for your recognition and positive feedback. We are pleased that our response was able to address your concerns.
> > >
> > > Thank you again for taking the time to engage with us, for your thoughtful feedback, and for raising your score for our paper from 4 to 5!

---

### Official Review · Reviewer_uRTr · 2026-03-13

**Soundness:** 3
**Presentation:** 3
**Significance:** 2
**Originality:** 2
**Overall Recommendation:** 5
**Confidence:** 2

**Summary:**

This paper introduces CoCLD (Coupled Continuous-Time Latent Dynamics), a framework for modeling irregular event sequences in continuous time by combining a time-guided diffusion module to recover latent states from sparse observations with a coupled Neural ODE that captures the joint evolution of individual behaviors and global dynamics, making the method particularly interesting for problems where local trajectories and population-level patterns influence each other over time.

**Compliance With Llm Reviewing Policy:**

Affirmed.

**Final Justification:**

The responses address my concerns well, especially by clarifying how the global state is constructed, how external factors are incorporated, how the framework is adapted across tasks, and what the practical efficiency looks like in comparison with relevant baselines.
These clarifications make the method more complete and convincing to me. While I still think some of these important implementation details should be stated more explicitly in the main paper, the rebuttal has resolved most of my uncertainty.
So I increase my score.

**Key Questions For Authors:**

Q1. Could the authors clarify how the global state is constructed in practice, including how the aggregation over individual latent states is implemented and how external factors are incorporated into the coupling process?

Q2. Could the authors provide a clearer task-specific description of how the generic CoCLD framework is adapted across different experimental settings?

Q3. Could the authors include a more complete efficiency evaluation of the full pipeline, such as training time, inference cost, memory usage, or parameter comparisons with strong baselines, to better demonstrate the practical overhead of the method?

**Limitations:**

Yes

**Strengths And Weaknesses:**

Strength 1
The paper addresses an important and practically relevant problem. It focuses on irregular and sparse event sequences, which are common in many real-world applications but remain challenging for existing models.

Strength 2
Instead of modeling only individual dynamics or incorporating global information in a shallow way, the paper explicitly captures the interaction between individual-level and population-level processes in a continuous-time framework, which makes the formulation more complete and better aligned with the underlying structure of many real-world dynamic systems.

Strength 3
The paper provides relatively solid theoretical support. Beyond the model design itself, the authors include theoretical analysis on properties such as existence, stability, and expressivity, which helps strengthen the paper by showing that the proposed coupled continuous-time system is not only empirically motivated but also supported from a mathematical perspective.

Weakness 1
The paper highlights the interaction between individual and population-level dynamics, but the global state is still described too abstractly. In particular, it is not very clear how the aggregation over individual latent states is implemented in practice, how much the results depend on this design, or how external factors are actually incorporated, which makes the coupling mechanism feel somewhat under-specified.

Weakness 2
The overall framework is interesting, but some implementation details are still not clear enough. Since the paper evaluates the method on multiple tasks with different settings, it would help to explain more explicitly how the generic CoCLD framework is adapted in each case; otherwise, it is hard to tell whether the gains mainly come from the proposed continuous-time dynamics model or from other architectural and training choices.

Weakness 3
The paper includes a complexity analysis, but the practical efficiency evaluation is still limited. Since the full pipeline combines latent interpolation, diffusion denoising, and coupled ODE modeling, it would be more convincing to report actual training time, inference cost, memory usage, or parameter comparisons with strong baselines, so that the real overhead of the method can be better understood.

---

> ### Author Rebuttal · Authors · 2026-03-29
>
> We greatly appreciate your thoughtful and encouraging comments, which have accurately captured the core motivations underlying our work. We are delighted that you acknowledge the significance of integrating individual- and population-level dynamics within a continuous-time framework, along with the rigor of our theoretical analysis.
>
> Below, we provide detailed clarifications regarding the implementation, task adaptations, and practical efficiency, which will be incorporated into the revised manuscript and appendix.
>
> **Response to [W1 & Q1]: Construction of the Global State, Aggregation, and External Factors.**
>
> We appreciate the opportunity to clarify these implementation details, which we will explicitly add to the methodology section in the revised manuscript:
>
> Aggregation over individual states ($\rho$): As briefly noted in the complexity analysis (Section 3.5), the aggregation function $\rho({h_{ind}^{(i)}(t)})$ is implemented using a mean-pooling operation across the latent states of all active entities within a specific time window or mini-batch. This design ensures that the global state efficiently captures the instantaneous macroscopic distribution without incurring quadratic computational costs, strictly maintaining the $O(M \times d)$ complexity.
>
> Incorporation of External Factors: External factors (such as time-of-day, day-of-the-week, or weather conditions, illustrated in Figure 1) are transformed into continuous-time embeddings. These context embeddings are directly concatenated with the aggregated global state $\mathbf{h}_{glob}(t)$ before being passed into the global vector field network $g\psi$. This explicitly conditions the population-level ODE on external environmental shifts.
>
> **Response to [W2 & Q2]: Task-Specific Adaptations of the Generic CoCLD Framework.**
>
> We want to emphasize that the core CoCLD architecture—*the Diffusion-based Latent Interpolator and the Coupled Neural ODEs*—remains completely identical and frozen across all three experimental settings. The performance gains stem fundamentally from our continuous-time dynamics model, not from task-specific. The generic framework is adapted only at the Input Encoder and Output Decoder levels (following Eq. 8).
>
> (1) Next-Event Prediction (Task 1): The Decoder is a standard Multi-Layer Perceptron (MLP) classification head over the discrete candidate space (e.g., locations). The model is optimized using Cross-Entropy loss.
>
> (2) Trajectory Generation (Task 2): The Decoder is adapted to output 2D continuous spatial coordinates (Latitude, Longitude). The model predicts the exact continuous location at target time $t_{target}$, optimized via Mean Squared Error (MSE) loss against real GPS points.
>
> (3) Sequential Behavior Modeling (Task 3): The Decoder computes dot-product similarity scores between the evolved coupled state $[h_{ind} \oplus Attn(h_{ind}, h_{glob})]$ and the learnable item embeddings, which is a standard setup in recommendation systems. It is optimized using Bayesian Personalized Ranking (BPR) loss.
>
> **Response to [W3 & Q3]:  Practical Efficiency Evaluation.**
>
> We appreciate this excellent suggestion. While CoCLD introduces coupled ODEs and diffusion, its practical overhead is highly manageable. We compared the empirical efficiency of CoCLD against a representative discrete baseline (SASRec), a continuous ODE baseline (SGODE), and strong diffusion baselines (DDRM and PreferDiff) on the IST dataset (using a single NVIDIA A800 GPU).
>
> | Model | Peak Memory (GPU GB) | Train Time (min/epoch) | Inference Time (ms/batch) |
> | :--- | :--- | :--- | :--- |
> | SASRec | 18 GB | 14 min | 68 ms |
> | SGODE |  53 GB | 36 min | 202 ms |
> | DDRM |  48 GB | 29 min | 127 ms |
> | PreferDiff |  35 GB | 24 min | 240 ms |
> | **CoCLD (Ours)** | 38 GB | 22 min | 110 ms  |
>
> As stated in Appendix E, we employ the Adjoint Sensitivity Method for the ODE solver. This keeps the memory cost of backpropagation $O(1)$ relative to the number of ODE integration steps ($N_{ODE}$), making our peak memory footprint (38 GB) comparable to other ODE models (e.g., SGODE with 53 GB) and much lower than heavy diffusion models (DDRM with 48GB).
>
> Inference Cost: PreferDiff relies on an iterative denoising process for auto-regressive generation, leading to high inference latency (240 ms). In contrast, CoCLD's *"Interpolate-then-Evolve"* paradigm only requires a single pass of the ODE solver during inference for temporal prediction, resulting in a much faster inference speed (110 ms), which is highly acceptable for real-world deployment.
>
> We hope these clarifications fully address your questions regarding the implementation and practical efficiency.
>
> If you find our responses satisfactory, we would deeply appreciate it if you could consider raising your score.

---

> > ### Author Rebuttal · Reviewer_uRTr · 2026-04-02
> >
> > Thank you for the rebuttal. The responses address my concerns well, especially by clarifying how the global state is constructed, how external factors are incorporated, how the framework is adapted across tasks, and what the practical efficiency looks like in comparison with relevant baselines. Overall, the rebuttal improves my assessment of the paper, and I will increase my score.

---

> > > ### Author Response · Authors · 2026-04-03
> > >
> > > Thank you so much for your positive acknowledgement and for recognizing that our rebuttal has addressed your concerns effectively.
> > >
> > > We are deeply grateful for your decision to raise your score from 4 to 5, which is a tremendous encouragement to our team.
> > > Thank you again for your time and valuable input.

---

### Official Review · Reviewer_UmC6 · 2026-03-13

**Soundness:** 2
**Presentation:** 3
**Significance:** 3
**Originality:** 3
**Overall Recommendation:** 4
**Confidence:** 4

**Summary:**

This paper studies how to model irregularly sampled event sequences in settings where both individual-level states and population-level dynamics evolve continuously over time and influence each other. The authors argue that existing methods either treat these two processes separately or rely on discrete-time approximations that fail under sparse, asynchronous observations. To address this, they propose CoCLD that jointly learns individual latent trajectories and global distributional shifts in a shared continuous-time latent space. The method combines a diffusion-based latent interpolator with Neural ODEs, enabling interpolation, generation, and alignment of latent states at arbitrary time points. The paper further claims that this coupling mechanism yields a consistent estimator of the underlying continuous-time latent dynamics under sparse and irregular observations. Experiments on next-event prediction, mobility trajectory generation, and sequential behavior modeling suggest that CoCLD captures dynamic dependencies effectively and outputs the baselines.

**Compliance With Llm Reviewing Policy:**

Affirmed.

**Final Justification:**

The authors addressed my main concerns.

**Key Questions For Authors:**

Please refer to the strengths and weaknesses above regarding the missing baselines. I would be willing to raise my score if these comparisons were added.

**Limitations:**

yes

**Strengths And Weaknesses:**

Strengths:
The paper is well structured and easy to follow. The experimental results are strong, showing that CoCLD consistently outperforms the baselines across three different types of tasks. In addition, the proposed method is theoretically grounded, which strengthens the overall contribution.

Weaknesses:
One important family of baselines is missing from the empirical evaluation: point process methods for irregular event modeling. In particular, the paper should compare against representative point process baselines, such as [1, 2, 3, 4], since these are highly relevant to the problem setting.

[1]  Mei, Hongyuan, and Jason M. Eisner. "The neural hawkes process: A neurally self-modulating multivariate point process." Advances in neural information processing systems 30 (2017).
[2] Omi, Takahiro, and Kazuyuki Aihara. "Fully neural network based model for general temporal point processes." Advances in neural information processing systems 32 (2019).
[3] Zuo, Simiao, et al. "Transformer hawkes process." International conference on machine learning. PMLR, 2020.
[4] Zhou, Zihao, and Rose Yu. "Automatic Integration for Fast and Interpretable Neural Point Processes." Learning for Dynamics and Control Conference. PMLR, 2023.

---

> ### Author Rebuttal · Authors · 2026-03-27
>
> We sincerely thank you for your feedback and for recognizing the clear structure, strong experimental results, and theoretical grounding of our paper. We also greatly appreciate you pointing out the missing family of baselines.
>
> We fully agree that **neural point process (NPP)** models for irregular event modeling are highly relevant to our problem setting and represent an important family of state-of-the-art baselines. Following your valuable suggestion, we have implemented the four representative point process models [1, 2, 3, 4] and added them to our empirical evaluation.
>
> **1. New Empirical Results**
>
> We conducted additional experiments on five datasets (e.g., IST, NYC, DC, Gowalla, and Brightkite) using the suggested baselines: Neural Hawkes Process (NHP)  [1], Fully Neural Point Process (FNPP) [2], Transformer Hawkes Process (THP) [3], and Automatic Integration for Fast and Interpretable Neural Point Processes (Auto-NPP) [4].
>
> As shown in the updated table below (which will be included in the final manuscript), the NPP baselines, particularly NHP and Auto-NPP, demonstrate strong performance. However, our proposed CoCLD consistently maintains state-of-the-art performance.
>
> |Dataset | Metrics | NHP | FNPP| THP | Auto-NPP | **CoCLD (Ours)** |
> | :---: | :---: | :---: | :---: | :---: | :---: | :---: |
> |**IST**| Acc@10 | 0.3568 | 0.2735 | 0.2950 | 0.3812 | **0.4255** |
> | | NDCG@10 | 0.2414 | 0.1750 | 0.2135 | 0.2704 | **0.2964** |
> |**NYC**| Acc@10 | 0.3972 | 0.2935 | 0.4113 | 0.4357 | **0.5083** |
> | | NDCG@10 | 0.2811 | 0.2192 | 0.2628 | 0.2991 | **0.3179** |
> |**DC**| Acc@10 | 0.3359 | 0.3051 | 0.3307 | 0.3437 | **0.3827** |
> | | NDCG@10 | 0.2366  | 0.2149 | 0.2413  | 0.2429 | **0.2620** |
> |**Gowalla**| Acc@10 | 0.3597  | 0.2750  | 0.2840   | 0.3511 | **0.3775** |
> | | NDCG@10 | 0.4426 | 0.3690 | 0.3712 | 0.4525 | **0.4725** |
> |**Brightkite**| Acc@10 | 0.3051 | 0.2906 | 0.3310 | 0.3413 | **0.3649** |
> | | NDCG@10 | 0.2677  | 0.2530 | 0.2918 | 0.3079 | **0.3245**  |
>
> **2. Analysis of the Results of our CoCLD and Neural Point Process (NPP) Baselines**
>
> While the suggested NPPs are elegant and powerful for modeling the conditional intensity of irregular events, CoCLD's superiority stems from addressing two specific challenges that standard NPPs do not fully resolve:
>
> *(1) Coupled Individual-Global Dynamics*: Models like NHP [1], THP [3], and Auto-NPP [4] primarily focus on modeling the historical dependency of a single sequence's intensity function. However, as discussed in our Introduction, real-world events are jointly driven by individual preferences and population-level distributional shifts (e.g., global mobility patterns, seasonal trends). CoCLD explicitly formalizes this mutual influence as a system of Coupled Neural ODEs (Eq. 3 & 4), ensuring that individual trajectories are continuously rectified by global distributions.
>
> *(2) Robustness to Extreme Temporal Sparsity*: Although FNPP [2] and Auto-NPP [4] solve the integration bottleneck to fit exact intensity functions, they are fundamentally forward-predictive models. When observations are extremely sparse (long missing intervals), their hidden states can suffer from "drift". In contrast, CoCLD introduces a Diffusion-based Latent Interpolator that reconstructs continuous latent paths probabilistically (Proposition 3.1). Our *"Interpolate-then-Evolve"* paradigm acts as a soft bridge over unobserved intervals, providing a more robust initial state for the continuous-time evolution.
>
> We believe that including these important baselines has significantly strengthened the comprehensiveness of our empirical evaluation.
>
> *We hope that these new results and discussions address your concerns, and we kindly ask you to consider raising your score as mentioned in your review.*
>
> Thank you again for your time and expertise!

---

> > ### Author Rebuttal · Reviewer_UmC6 · 2026-04-04
> >
> > Appreciate the rebuttal. I have raised my score accordingly.

---

> > > ### Author Response · Authors · 2026-04-04
> > >
> > > Thank you very much for your careful review and positive feedback. We greatly appreciate your time, effort, and constructive comments that helped us improve the paper, and we are grateful for your updated score.

---

### Decision · Program_Chairs · 2026-04-30

**Decision:**

Accept (spotlight)

**Comment:**

The reviewers agree that this is an interesting, useful and well-executed paper. The reviewers had a few concerns about the presentation (issues like the global state, algorithmic details, and computational complexity) as well as empirical evaluation. These were satisfactorily addressed by the authors in their responses: please go again over the reviews as well as your rebuttal while revising the paper.